# Transcriptomes of Injured Lamprey Axon Tips: Single-Cell RNA-Seq Suggests Differential Involvement of MAPK Signaling Pathways in Axon Retraction and Regeneration after Spinal Cord Injury

**DOI:** 10.3390/cells11152320

**Published:** 2022-07-27

**Authors:** Li-Qing Jin, Yan Zhou, Yue-Sheng Li, Guixin Zhang, Jianli Hu, Michael E. Selzer

**Affiliations:** 1Shriners Hospitals Pediatric Research Center, The Lewis Katz School of Medicine (LKSOM) at Temple University, Philadelphia, PA 19140, USA; kzhang59@temple.edu (G.Z.); jianli.hu@temple.edu (J.H.); 2Department of Neural Sciences, Lewis Katz School of Medicine (LKSOM), 3500 North Broad Street, Philadelphia, PA 19140, USA; 3Biostatistics and Bioinformatics Facility, Fox Chase Cancer Center, Philadelphia, PA 19111, USA; yan.zhou@fccc.edu; 4DNA Sequence & Genomics Core Facility at the NHLBI, Bethesda, MD 20892, USA; yuesheng.li@nih.gov; 5Department of Neurology, Lewis Katz School of Medicine (LKSOM), 3500 North Broad Street, Philadelphia, PA 19140, USA

**Keywords:** spinal cord injury, single cell RNA-seq, axon regeneration, MAPK pathway, circadian rhythm, local protein synthesis

## Abstract

Axotomy in the CNS activates retrograde signals that can trigger regeneration or cell death. Whether these outcomes use different injury signals is not known. Local protein synthesis in axon tips plays an important role in axon retraction and regeneration. Microarray and RNA-seq studies on cultured mammalian embryonic or early postnatal peripheral neurons showed that axon growth cones contain hundreds to thousands of mRNAs. In the lamprey, identified reticulospinal neurons vary in the probability that their axons will regenerate after axotomy. The bad regenerators undergo early severe axon retraction and very delayed apoptosis. We micro-aspirated axoplasms from 10 growing, 9 static and 5 retracting axon tips of spinal cord transected lampreys and performed single-cell RNA-seq, analyzing the results bioinformatically. Genes were identified that were upregulated selectively in growing (*n* = 38), static (20) or retracting tips (18). Among them, *map3k2*, *csnk1e* and *gtf2h* were expressed in growing tips, *mapk8(1)* was expressed in static tips and *prkcq* was expressed in retracting tips. Venn diagrams revealed more than 40 components of MAPK signaling pathways, including *jnk* and *p38* isoforms, which were differentially distributed in growing, static and/or retracting tips. Real-time q-PCR and immunohistochemistry verified the colocalization of *map3k2* and *csnk1e* in growing axon tips. Thus, differentially regulated MAPK and circadian rhythm signaling pathways may be involved in activating either programs for axon regeneration or axon retraction and apoptosis.

## 1. Introduction

When an axon is severed, e.g., due to spinal cord injury (SCI), its cell body responds in one of two ways. In some neurons, a regenerative response is activated, which allows the axon to regrow. In other cases, the neuron undergoes retrograde death or atrophy. Several retrograde signals have been implicated in these injury responses, including members of the mitogen-activated protein kinase (MAPK) family [1,2,3]. So far, it is not known whether the divergent responses to injury (regenerative vs. degenerative) are caused by different or the same retrograde signals. In the mammalian central nervous system (CNS), repair mechanisms are inadequate to reverse the functional impairment, and there is no effective treatment available for SCI. It is widely accepted that intra-axonal protein synthesis and degradation play an important role in axon regeneration [4,5,6,7], allowing the growing axon tip to respond rapidly to its needs despite its long distance from the perikaryon. Local translation occurs in axons of the injured peripheral nervous system (PNS) [8], in growth cones during early development in vitro [9,10,11] and in vivo [12]. Using microarrays, up to two thousand mRNAs have been identified in cell cultures, including dorsal root ganglion cells (DRG), cortical neurons and sympathetic neurons (reviewed in [13]). More than 6000 mRNAs were reported by deep sequencing (RNA-seq) in axons of explanted embryonic mouse sensory neurons [14]. The interpretations of the results in the above studies were based mostly on whether the axon had already regenerated, or what stage of development it was in, rather than on its behavior at the time of assaying. Sampling RNA from axon tips is difficult, due to the complexity and opacity of the mammalian CNS. We have not found reports of RNA-seq from individual axon tips in the injured mammalian spinal cord. In contrast to mammals, lampreys, the most primitive extant vertebrates, regenerate supraspinal axons spontaneously after complete spinal cord transection (SC-TX), leading to the recovery of almost normal-appearing locomotion [15,16,17,18,19]. The sea lamprey brain has approximately 2000 spinal-projecting neurons. Among them, 36 are individually identifiable. Their large cell bodies and growing tips (~150 µm), as well as the translucency of the spinal cord, make it feasible to image the injured axon tips live and to micro-aspirate their axoplasm. The proximal stump of axons seal and retract as much as several mm during the first few days. Axons in the distal stump undergo Wallerian degeneration, and after 1–2 weeks, the sealed axon tips begin to regenerate in the proximal stump. Axonal regeneration in lamprey spinal cords is robust but not complete. About half of the spinal-projecting axons regenerate beyond the lesion by 12 weeks post-transection [20,21] but only a few mm beyond the lesion rather than all the way back to their previous caudal-most targets [15,17,20], although longer distances have been described at very long recovery times [22]. Of special interest, the identified reticulospinal neurons vary widely in their response to axotomy [21,22]. Those that are poor regenerators (i.e., the probability that their axons will regenerate beyond the lesion is low) show the greatest amount of early retraction and are most likely to undergo very delayed apoptosis [23,24,25,26,27]. By reimaging living axons in exposed spinal cords over a period of 2–4 h and locating the positions of the axons relative to fiduciary landmarks, we have been able to determine the growth status (growing, static or retracting) of individual axon tips [28,29] at the time of micro-aspiration. As in peripheral nerve and neuronal cultures, the injured axon tips of lamprey spinal cords contain mRNAs and polyribosomes, suggesting that axon regeneration may involve local protein synthesis [29].

Here, we have combined micro-aspiration and the isolation of total RNA from individual axon tips with deep sequencing to gain insight into the role of local protein synthesis in selecting regenerative vs. degenerative responses to axotomy in the lamprey spinal cord. The results support differential roles for different MAPK signals in axon regeneration and retraction.

## 2. Materials and Methods

### 2.1. Spinal Cord Transection and Retrograde Labeling of Axons

A total of 260 larval lampreys (*Petromyzon marinus*), 10–13 cm in length (4–5 years old and in a stable phase of neurological development), were used for the current research. They were either purchased from Lamprey Services (Ludington, MI, USA) in Michigan or gifted (100) by Dr. Nicholas Johnson of the USGS Great Lakes Science Center (Cleveland, OH, USA) and maintained with light cycles in freshwater tanks at 15 °C until use. Among them, 200 animals were used for the micro-aspiration in practice and in formal experiments. The rest were used for histological studies. Lampreys were anesthetized by immersion in saturated aqueous benzocaine for 5 min and were pinned to a Sylgard (184 silicone elastomer, Dow Corning) plate filled with ice-cold lamprey Ringer (110 mM NaCl, 2.1 mM KCl, 2.6 mM CaCl_2_, 1.8 mM MgCl_2_ and 10 mM Tris buffer; pH 7.4). The spinal cord was exposed via a dorsal incision and was transected under direct microscopic vision with iridectomy scissors at the level of the 7th gill. Spinal axons and their cell bodies in brainstems were retrogradely labeled with the fluorescent dye dextran-tetramethylrhodamine (DTMR, 10 kDa, 5% in 0.1 M Tris buffer, pH 7.4) at the time of SC-TX by inserting a 0.5 mm dye-soaked Gelfoam pledget into the TX site. The cut axons and their cell bodies became brightly labeled within several hours. The Gelfoam was no longer distinguishable from the overlying scar tissue by 10 days after insertion. Background fluorescence was largely cleared from the area of dye application by 1 week after application. Although a fluorescent label was picked up by glial cells and small neurons in the region adjacent to the application, the resulting low level of fluorescence did not interfere with the observation of the large RAs and their cut tips. Animals were covered with a lamprey Ringer-moistened paper towel and were kept on ice for 3 h to facilitate clot formation. Animals were returned to fresh water and were allowed to recover at 4 °C overnight, and then they were housed at room temperature.

### 2.2. Determination of Axon Tip Growth Status

Lampreys were re-anesthetized, and their cords were re-exposed after 10 to 14 days of recovery. The spinal cord was exposed by a dorsal midline incision from the level of the third gill to the level of the original TX, a distance of approximately 8 mm. DTMR-labeled axon tips were visualized, and the first image was captured under a fluorescence dissecting microscope (Olympus SZX-TBI, Tokyo, Japan) with a CCD camera (Olympus DP72, Tokyo, Japan) and software (cellSens, version 1.16) without lamprey Ringer. Fiduciary landmarks such as surface capillaries and local neuronal cell bodies were noted and used for later alignment. Then, the spinal cord was covered with a Gelfoam pledget pre-soaked with lamprey serum, and its head and body were covered with Kimwipes pre-soaked with lamprey Ringer. The animals were kept in a cold, high humid environment (a Coleman Cooler with ice) for 3–4 h, and their spinal cords were reimaged after 2–4 h. The growth status of a given axon tip (growing, static or retracting) during that interval was determined by measuring the change in the location of the axon tip relative to the fiduciary landmarks. To avoid bias in finding the front edge, images were converted to gray mode and were processed using the tool “Find Edges” (Adobe Photoshop CS4, Version: 11.0.1). This allowed us to discriminate distances as short as 2–3 µm between observations. A tip was deemed “growing” or “retracting” if it had moved more than 10 µm towards or away from the transection site. The rest were judged “static tips”. Axon tips that had moved between 3 and 10 µm were excluded to avoid ambiguity. Of 33 microaspirated axoplasms, 9 were excluded, either because of this ambiguity, because the leading edge of the tip could not be identified unambiguously in the fluorescence micrographs or because the tip had to be impaled a second time in order to microaspirate sufficient axoplasm, which may have introduced contaminating material from surrounding cells (Appendix A). Their raw data are included in Appendix A.

### 2.3. Isolation of the Axoplasmic Contents from Individual Axon Tips

The lamprey spinal cord is flat and translucent, allowing the visualization of cells and fluorescently labeled axon tips in living preparations [24,28]. The largest of these axons are several pairs of Müller axons in the ventral columns and the pair of Mauthner axons in the lateral axon columns. These all have been traced to the perikarya of large individually identified reticulospinal neurons by serially sectioning through the entire nervous system in a single specimen [30]. With the exception of the Mauthner axon, the degree of variability in the arrangement of these large axons within the axon tracts is still not known. Moreover, the narrowing of the axon at the initial segment may also make it difficult to trace even fluorescently labeled axons all the way to their perikaryon in CNS wholemounts. Thus, we did not attempt to identify specific individual axons. However, the rate of elongation of a regenerating axon could be discerned in 2–4 h [28]. After axotomy, the sealed proximal tips form spindle-shaped or oval structures, which can be as large as 300 (length) × 100 (diameter) µm. This enabled penetration and micro-aspiration through glass micropipettes pulled from 1.5~1.8 mm o.d. microcapillary tubing (KIMAX 51, Kimble, Vineland, NJ, USA), using a Narishige PC-10 puller, and kept in a clean environment before use. Micropipettes were advanced with a Leitz micromanipulator under stereoscopic fluorescence microscopy at 63× magnification, with 8 psi positive pressure applied to prevent the entry of contaminants. When a slight blanching of fluorescence indicated dimpling of the axon tip, positive pressure was replaced by negative pressure, and the micropipette tip was advanced further. Successfully penetrated axon tips collapsed, and the fluorescence was displaced into the micropipette tip, which was quickly withdrawn. The contents (~0.1 µL) were carefully expelled into an autoclaved 0.65 mL microcentrifuge tube with 3 µL of RNase inhibitor (RNaseOUT, Invitrogen, Carlsbad, CA, USA). Samples were frozen immediately in dry ice and were kept at −80 °C until use.

### 2.4. Construction of Libraries

Two kits were used in constructing the libraries: (1) SMART-Seq^®^ Single Cell Kit (Takara, San Jose, CA, USA; Cat. 634471) and (2) Nextera XT DNA Library Prep Kit (Illumina, San Diego, CA, USA; Cat. FC-131-1024)/IDT^®^ for Illumina^®^ DNA/RNA UD Indexes Set A, Tagmentation (Illumina, Cat. 20027213). Briefly, a 2 µL aliquot of axoplasm from an individual axon tip was incubated with RNase inhibitor in 1 × reaction buffer with 3′ SMART-Seq CDS Primer II A at 72° for 3 min, and it was reacted with RT Master Mix for 180 min at 42 °C to synthesize first-strand cDNA, which was amplified by LD-PCR in SeqAmp CB PCR buffer, containing PCR primer and SeqAmp DNA polymerase (98 °C for 10 s, 65 °C for 30 s, 68 °C for 3 min × 20 cycles). The quality of the cDNA was checked by a 2100 Bioanalyzer (Agilent Technologies, Santa Clara, CA, USA). NGS libraries were then constructed with the Nextera XT DNA Library Prep kit per Illumina’s instructions. The quality of DNAs was rechecked, and the DNA was quantified and pooled to generate a final concentration of ~5 nM in 120 µL.

### 2.5. scRNA-seq and Data Analysis

The fragment sizes of libraries were verified using the Agilent 2100 Bioanalyzer, and the concentration was determined using a Qubit instrument (LifeTech, Carlsbad, CA, USA). The libraries were loaded onto Illumina Novaseq 6000 for 2 × 100 bp paired-end read sequencing. FASTQ files were generated using the bcl2fastq software for further analysis. Sequence reads were aligned to the curated Lamprey genome (Pmarinus_7.0, version 105) using STAR [31]. The number of raw counts in each known gene from the RefSeq database was enumerated using htseq-count from the HTSeq package [32]. Differential expression between samples and across different conditions was assessed for statistical significance using the R/Bioconductor package DESeq2 [33]. Genes with a false discovery rate (FDR) ≤ 0.2 and a fold-change (FC) ≥ 1.5 were considered significant. Enrichment analyses were performed in the Database for Annotation, Visualization and Integrated Discovery (DAVID) Bioinformatics Resources platform [34]. Gene symbols were converted to Entrez-ID by GeneALaCart (https://genealacart.genecards.org/, accessed on 26 July 2022) and were uploaded onto the DAVID internet platform. After “Fold Enrichment” and “FDR” options were checked, the “Functional Annotation Chart” module was re-calculated. Results were downloaded for further analysis in an excel file. Principal component analysis (PCA) and regression analysis were performed using the R/Bioconductor DESeq2 package. For protein interaction analysis, 2 original networks were generated by searching the PubMed database with “*map2k3*” and “*csnk1e*” in Cytoscape (v. 3.9.0) software, analyzed and trimmed by the Search Tool for the Retrieval of Interacting Genes/Proteins (STRING) [35] and DAVID databases and edited using Cytoscape and Adobe Photoshop (CS4) software.

### 2.6. Cryostat Sectioning

The length of the body containing the brain and spinal cord rostral to the TX site was obtained by cutting the frozen tissue in the olfactory sac in the head and 1 mm below the level of the 7th gill in the spinal cord. The head was separated from the spinal cord by a third cut at the 1st gill, and the spinal cord stump that contained the TX site was obtained by a 4th cut at the 3rd gill. The frozen stumps were placed in stainless steel molds (10 × 10 × 10 mm, Simport Scientific, Saint-Mathieu-de-Beloeil, QC, Canada) filled with Optimal Cutting Temperature (OCT) medium. The orientation of the frozen tissue was adjusted to obtain transverse or coronal (horizontal) sections. The molds were placed directly on top of liquid nitrogen in a Dewar flask until the OCT became completely solid. Blocks were stored at −80 °C until sectioning. Cryosectioning was performed on a Leica CM1950 Cryostat with the chamber temperature at −17 °C (Leica, Wetzlar, Germany). The frozen blocks were placed in a cryochamber for 30 min to achieve uniform temperature before sectioning. The OCT-embedded specimens were cut serially at 15 µm and were mounted on glass slides. The slides were kept temporarily in a Styrofoam box filled with dry ice during sectioning and were then stored at −80 °C until use.

### 2.7. Immunohistochemistry (IHC)

For IHC, slides were thawed and dried in a plastic box filled with Drierite desiccant (Drierite Co, Xenia, OH, USA) at 4 °C for 15 min and then at room temperature for 30 min. The tissue was fixed with ice-cold acetone for 2 min and was washed with phosphate-buffered saline (PBS) twice for 30 s to remove OCT and the fluorescent dye DTMR. Nonspecific binding was blocked with 4% fetal bovine serum (FBS, in PBS, pH 7.5) for 5 min. Slides were incubated with antibodies against map3k2 (Cell Signaling, Beverly, MA, USA; #19607) and csnk1e (DSHB, Iowa, IA, USA; AFFN-CSNK1E-13D7) overnight at 4 °C. The next day, samples were washed in Tris-buffered saline solution (TBS, 0.1% Tween 20 in PBS, pH 7.5) and were blocked by normal animal serums depending on the origin of the secondary antibodies. For immunofluorescence, slides were incubated with either Alexa Fluor 488 donkey anti-mouse IgG polyclonal antibodies (Invitrogen, Carlsbad, CA, USA; Cat# A11029) or Alexa Fluor 594 donkey anti-rabbit IgG polyclonal antibodies (Biolegend, San Diego, CA, USA; Cat# 406418) at 1:200 in blocking solution for 1 h at room temperature. Finally, the specimens were washed twice with TBS and were mounted with DAPI Fluoromount-G (Southern Biotech, Birmingham, UK; 0100-20). Images were captured using a Nikon Eclipse 80i epifluorescence microscope with a Roper Scientific CoolSNAP digital camera (Trenton, NJ, USA), under the control of NIS Elements software. Negative controls employed the same protocol, omitting the primary antibody. For chromogenic IHC, protocols recommended by Cell Signaling Technology were followed. It generated brown to gray–black colors.

### 2.8. Real-Time Quantitative Polymerase Chain Reaction (q-PCR)

Pre-amplified tip cDNAs (20 cycles) by SMART-Seq^®^ Single Cell Kit in the “Construction of libraries for scRNA-seq and sequencing” were used to perform the q-PCR with Power SYBR Green Master Mix (Thermo Fisher, Cat. 4368577, Waltham, MA, USA) on a qPCR apparatus (StepOnePlus Real-Time PCR system thermal cycler, Applied Biosystems, Waltham, MA, USA). Real-time q-PCR was performed in triplicate on 96-well reaction plates under the following conditions: activation (95 °C ×10 min), PCR 40 cycles (95 °C ×15 s, 60 °C ×1 min) and data collection (95 °C-10 min, 60 °C-1 min), followed by cooling to 40 °C. The primers used in q-PCR are presented in Table 1. For relative quantification, the expression levels of genes in the growing tips or retracting tips were normalized to those from static tips.

### 2.9. Western Blotting

A total of 10 *Petromyzon marinus* specimens were anesthetized in saturated aqueous benzocaine, and their brains and spinal cords were dissected. Tissues were homogenized in RIPA buffer (Pierce, Appleton, WI, USA) containing 20 mM Tris-HCl (pH 7.5), 150 mM NaCl, 1 mM Na_2_EDTA, 1 mM EGTA, 1% NP-40, 1% sodium deoxycholate, 2.5 mM sodium pyrophosphate, 1 mM β-glycerophosphate, 1 mM Na_3_VO_4_ and 1 µg/mL leupeptin, by sonication (FB120 Sonic Dismembrator, Fisher Scientific, Waltham, MA, USA). The homogenate was centrifuged at 10,000× *g* for 20 min. The supernatant was collected and used. Their protein concentrations were quantified by the Lowry method. Proteins (35 µg) were separated in a 10% SDS mini-gel and were electrophoretically transferred onto nitrocellulose membranes (0.45 µm, Bio-Rad, Philadelphia, PA, USA) using a Bio-Rad transblot apparatus. The membranes were blocked in an Odyssey Blocking Buffer (LI-COR, 927-40000) for 2 h and were incubated with antibodies against map3k2 (Cell Signaling, #19607) and csnk1e (DSHB, AFFN-CSNK1E-13D7) overnight at 4 °C, followed by washing with PBS containing 0.2% Tween 20 (PBST) and by incubation with the fluorescence-labeled secondary antibody IRDye 680RD (926-68071, LI-COR) for map3k2 and IRDye 680RD (926-68070, LI-COR) for csnk1e for 1 h at room temperature. The membranes were then rinsed in PBST and were scanned by the Odyssey infrared imaging system (LI-COR Biosciences, Lincoln, NE, USA).

### 2.10. Statistical Analysis

All data are represented as the mean ± standard error for all performed repetitions. Statistically significant differences among three or more groups were analyzed by a one-way analysis of variance (ANOVA), followed by Tukey’s post-hoc analysis. The significance level was set to *p* < 0.05. The statistical analysis was calculated through SigmaStat software for Windows (version 2.0). Sample numbers are indicated in the figure legends.

## 3. Results

### 3.1. Determination of Growth Status and the Micro-Aspiration of Axoplasms

A total of 80 animals were used in sample collection by micro-aspiration, as described in the Section 2. In approximately half of these, either we could not find any giant axon tips, or their leading edges were not easily distinguishable at the time of aspiration. The success rate of micro-aspiration was 60–70%. Thus, 33 axoplasms were micro-aspirated in 30 animals (body length = 9.4 ± 0.1 cm, mean ± SEM). These axoplasms were further processed to synthesize first-strand cDNAs for initial amplification. After electrode withdrawal from the spinal cord, axoplasm was checked under the fluorescence dissecting microscope with constant light intensity. All aspirates generated bright fluorescence images with exposure times of less than 5 s (insets in Figure 1). Successful aspirations collected about 0.095–0.105 µL, measured by isotope scintillation counting, and they contained 1–10 pg of total RNA, assessed by comparison with the neurofilament signals in control total lamprey CNS RNA [29]. Positive pressure was supplied to the electrode to expel the aspirant into 3 µL of reaction buffer containing RNase inhibitor. During this process, only the aspirant droplet was directly in contact with and entered the reaction buffer, avoiding any contact between the exterior wall of the glass electrode and the reaction buffer solution. Among 33 samples, 16 were from growing tips (G tips; elongation by +27.9 µm ± 3.5), 11 were from static tips (S tips; 0.0 µm ± 0.1) and 6 were from retracting tips (R tips; −21.3 µm ± 3.1) within approximately four hours of observation (4.2 h ± 0.1, *n* = 30). Representative examples of these three types of tips are shown in Figure 1.

### 3.2. Construction of scRNA-seq Libraries

Thirty-three RNA-seq libraries were constructed with the SMART-Seq Single Cell Kit and Nextera XT DNA Library Prep kit. Due to the limited amount of total RNA (1–10 pg in 0.1 µL) in each aspiration, it was not practical to measure the quality of mRNAs before PCR amplification. Therefore, we postponed the quality assessment until the mRNAs were reverse-transcribed and amplified by the first round of PCR amplification (×20 cycles, the 1st kit) and until after tagmentation, followed by another around of PCR amplification (×12 cycles, the 2nd kit). An amount of 2 µL of each PCR product was checked for the quality of synthesized transcriptomes. The top image in the right panel of Figure 2 shows the DNA assayed with a 2100 Bioanalyzer from a sample before tagmentation, with a fluorescence intensity curve covering 0.5–2.5 kb and a peak at approximately 1.68 kb. The bottom image shows the result from another sample after tagmentation and PCR amplification. The curve covers 0.15–1.95 kb with a peak at approximately 735 bp. The single principal peak flanked by flat baselines indicates that the transcriptome quality was satisfactory. The smaller cDNA sizes of the post-tagmentation fluorescence peak compared to those in the pre-tagmentation peak implies that the tagmentation of cDNAs was catalyzed efficiently.

### 3.3. RNA-Seq, Alignments and Assemblies

FASTQ files (100+ Gb), generated using the bcl2fastq software, were analyzed to identify differentially expressed genes (DEGs). Data trimming and alignment were processed as described in the Section 2. Raw counts for each sample were obtained for 33 transcriptomes (Appendix A). Samples were further selected based on two criteria: (1) to avoid possible contamination of the aspirant with material from surrounding cells, only one penetration per tip was allowed; (2) axon tips were excluded if, after careful examination of fluorescence images, the growth status of an axon tip was ambiguous because its leading edge could not be identified (Appendix A), as described in the Section 2. This left 24 samples (G = 10, S = 9, and R = 5) for inclusion in the differential gene expression analysis.

### 3.4. DEG Identification and DAVID Enrichment Analysis

Differential expression between G tips, S tips and R tips was analyzed using the DESeq2 package, and FCs and adjusted *p*-values (Padj, FDR) were calculated for each gene in three comparisons (G vs. R; G vs. S; and S vs. R). Only genes with FC ≥ 1.5 and FDR ≤ 0.2 were included in subsequent hierarchical cluster analyses. This threshold resulted in three lists of genes that were differentially expressed between G tips and R tips (group A = 1488 genes, Appendix A), G tips and S tips (group B = 151, Appendix A) and S tips and R tips (group C = 1067, Appendix A).

Based on similarities in gene expression patterns, the genes in these three groups were classified in expression clusters by the hierarchical algorithm (Appendix A). These clusters contained genes that were up- or downregulated in the axon tips of each growth status compared to the other two statuses. The heat maps in Figure 3 show three representative clusters that were upregulated in G tips compared with S tips (Figure 3A) or R tips (Figure 3B), or that were downregulated compared with R tips (Figure 3C). Figure 3D consists of three lists of genes: those from groups A and B that were consistently expressed at higher levels in G than in S or R tips, respectively (G > S and R, 38 genes); those from groups B and C that were expressed more in S than in G or R tips, respectively (S > G and R, 20 genes); and those from groups A and C that were expressed more in R than in G or S tips, respectively (R > G and S, 18 genes). Briefly, a subgroup of genes from Appendix A (G > R) and Appendix A (G > S) with positive *log2FoldChange* values represent genes upregulated in G vs. R and G vs. S comparisons. The shared genes in the two subgroups form the G > S and R list in Figure 3D. Each gene in the above three lists was analyzed for possible involvement in signaling pathways by three default pathway databases: BBID, BIOCARTA and KEGG (Appendix A). BBID reported only the MAPK signaling pathway represented by *map3k2* and *mapk8*. BIOCARTA, a cell signaling pathway database [36], presented four pathways: the circadian rhythm (*csnk1e*), MAPK (*map3k2*, *mapk8*), sonic hedgehog (*smo*) and keratinocyte differentiation (*prkcq*) pathways. KEGG generated more than 40 pathways related to 26 genes, including the 5 genes mentioned above (Appendix A). It is well-known that kinases play important roles in cellular functions by activating proteins. Therefore, we used BIOCARTA, which is most likely the most frequently relied upon. Kinases are the second-most drug-targeted proteins after the G-protein coupled receptors [37]. Among the three lists of genes, only four kinases were found, encoded by genes *csnk1e*, *map3k2*, *mapk8(1)* and *pckcq*. These are presented in red, green, blue and purple, respectively, for their upregulation in G, S and R tips. As a result of the space limitation, and because the genes upregulated in G tips are of greater priority in the present context, we especially focused on *map3k2*, *csnk1e* and their related pathways, on MAPK and the circadian rhythm pathways. Although MAPK signaling pathways have been studied for many years with regard to their roles in axon regeneration, their true effects remain unclear due to the complexities associated with the co-existence of Erk, Jnk, p38 cascades and the cross-interactions among them. Circadian timing has been implicated in the regulation of several cell functions linked to tissue regeneration via the homeostasis of stem cell proliferation [38]. However, a role in axon regeneration has not yet been demonstrated. Two volcano maps showing both *map3k2* and *csnk1e* are found in the right side (upregulated) of the left (G vs. R) and right (G vs. S) panels of Figure 4.

From the three groups, A, B and C, we also identified 60 genes, the top 10 of each of the 6 possible categories (G > R, R > G, G > S, S > G, S > R, and R > S) in FC and −log_10_FDR values (Table 2, Table 3 and Table 4), with their GO terms (GeneCards). Among the 10 genes of the category G > R, 3 are involved in RNA transcription and RNA binding (*leo1*, *actn4*, *mrpl12*), 1 is involved in protein synthesis (*eif2s1a*), 3 are involved in mitochondrial function (*alas2*, *sdha* and *mrpl12*), 1 is involved in chromosome regulation (*gpatch11*), 1 is involved in the heat shock protein of the Hsp70 family (*dnaja4*) and 1 is involved in phosphatidylinositol glycan biosynthesis (*pigq*). The biological functions of the 10 genes in the category R > G include involvement in lymphocyte signaling and differentiation (*btk, prkcq*), the inhibition of cAMP accumulation (*rxfp3*) and protein degradation (*mul1a*). In another category (G > S), genes were identified that are involved in the *ras*-related signaling pathway (*stk19*, a regulator of N-Ras signaling) or in the structural maintenance of chromosomes (*smchd1*). In contrast, genes involved in exocytosis (*scamp5*), the extracellular matrix (*tectb*) or insulin-like growth factor (IGF) binding (*wisp2*) were found in the S > G category. Interestingly, the Erk/MAPK-signaling-pathway-related gene *raf1a* was found among the top 10 of the S > R category. Therefore, a ras-raf-MAPK signaling pathway may be involved in static tips to maintain axon stability. In the R > S group, *sdc2* and *prkcq* are genes related to apoptosis.

The DAVID Bioinformatics Resources program was used for the functional analysis of transcripts [34]. Three groups of differentially expressed genes (DEGs) were analyzed and sorted. Their GO terms are presented in Figure 5A, Figure 6A and Appendix A or in Figure 5B, Figure 6B and Appendix A (FC > 1.5 and *p*-value < 0.05). The most significant GO terms (FDR < 0.05) were found in group A genes (G tips vs. R tips, Figure 5A). Two more general cell functions were found: protein synthesis (red GO terms) and mitochondrial functions (green GO terms). The former includes GO terms of r-, t- and m-RNA processing; poly(A) RNA binding; ribosome biogenesis; and protein biogenesis. The latter includes mitochondrial and iron-containing proteins. Genes under important GO terms are presented in (Appendix A). Three GO terms that were not significant by FDR criteria in Figure 5A (Mitosis, Exosome, Exonuclease, 0.1 < FDR < 0.05) were significant by *p*-value criteria (*p* < 0.05) and are included in Figure 5B, reflecting the higher stringency of FDR compared with *p*-values.

Compared with group A (G tips vs. R tips, Figure 5A), group C (S tips vs. R tips, Figure 6A) generated few significant GO terms, suggesting that S tips may have lower levels of local protein synthesis and mitochondrial function than G tips, and they have expression patterns close to R tips. The enrichment analysis of group B genes (G tips vs. S tips) generated fewer GO terms (Appendix A), implying a high degree of similarity between the G tips and S tips in their expression levels and patterns. Interestingly, the “MAPK-pathway” (BBID) was identified (FC = 17.6, FDR = *p*-value = 0.057, Appendix A). No significant GO terms were found from a list of 60 genes that are upregulated in R tips vs. S tips (FDR < 0.05), although 2 genes, *fnta* (protein farnesyltransferase) and *prkcq* (PKC-θ), are listed in one GO term: “cellular component disassembly involved in execution phase of apoptosis” (*p*-value = 0.048).

The different FDR values used above were selected to accomplish different goals. For DEG analysis, a low threshold (FDR ≤ 0.2) was selected in order to increase the chances of identifying genes whose encoded proteins are related to epigenetic modifications, or that are involved in signaling pathways, because these genes and coded proteins are more likely to be differentially expressed in different growth states, even if their expression levels are low. This increased the number of genes that could be analyzed by gene ontology (GO) enrichment, whereas a higher stringency criterion (FDR < 0.05) was used in most cases in order to exclude noisy signals. The low-threshold FDR did not affect the identification of *map3k2* as a gene present selectively in growing axon tips, nor did it affect other analyses, such as the Venn diagram analysis or the analyses of protein–protein interactions (PPI). GO terms with higher FDR values (FDR < 0.1 in Figure 5A and Appendix A; FDR < 0.2 in Figure 6A) were indicated and included only for comparison with corresponding scatter plots in Figure 5B, Appendix A and Figure 6B, respectively, where *p* values < 0.05 were used.

To assess the variation of whole samples, we performed PCA on all sequencing data from 24 transcriptomes (cutoff > 5, Appendix A). The results indicated that the expression patterns of the three groups of tips (G, S and R) were indiscernible. Most tips were clustered together on the top-left of the PCA diagram. The main source of variance came from two tip samples, G4 and S2 (Appendix A). Two lists of genes (143, 181) were found to be expressed at very high levels in G4 and S2, respectively. No significant GO terms were found (FDR > 0.43, 0.22, for G4 and S2 lists, respectively) by enrichment analysis.

### 3.5. Venn Diagram Analysis of Genes Participating in MAPK Pathways and Regulating Histone Functions

Most of the transcripts in the G, S and R tips did not show differential expression patterns when studied by DESeq2. Therefore, we performed a Venn diagram analysis for all the transcripts identified from the three individual groups (G, S and R tips). Since the numbers of tips in each group differed, which can result in bias toward the selection of more genes in the larger groups if all samples were used, we selected the five most related transcriptomes from each group (G, S, R) to participate in the Venn diagram analysis, based on a Pearson correlation analysis of counts within each group. For example, in group G, the coefficients of determination (R^2^s) were obtained from paired counts from G1-G2, G1-G3, …, G1-G10, G2-G3, etc. Sums of R^2^s for each transcriptome (G1, G2, …, G10) were calculated (Appendix A). Consistent with PCA analysis, two axon tips (G4 and S2) having the lowest R^2^ values were excluded. From the G and S groups each, the five members with the highest R^2^ values—G5-7, G9-10, S5-9—were selected, together with R1-5, to generate three merged lists of genes (cutoff > 1), representing the repertoires of RNA-seq in G, S and R axon tips. A Venn diagram was plotted (BioVenn) with areas proportional to the number of gene species (Figure 7). There were more transcripts in G tips (5097) than S tips (4105) or R tips (2318). The diagram subdivides these three lists of genes into seven divisions in Venn diagram: v-GSR (1431), v-GS (1650), v-GR (384), v-SR (187), v-G (1632), v-S (837) and v-R (316).

The results of the above analyses suggest that MAPK pathways may be involved in axon regeneration, as shown in Figure 3D (G > S and R, S > G and R) and Appendix A. We conducted an analysis of the distributions in seven Venn diagram divisions of transcripts related to the MAPK pathway. GO enrichment analyses were performed to identify related transcripts. Out of 253 listed MAPK genes (DAVID Gene Names), 45 (17.8%) were found in axon tips (Figure 7). A total of 3 conventional MAPKs were unevenly distributed in the diagram: *erk*, *jnk1*, *p38α* in v-GS, *jnk2* in v-R and *p38β* in v-GR, implying their different function in axon regeneration. GO enrichment analysis suggested that genes related to histone function regulation (the blue GO terms in Figure 5) played important roles in axon regeneration, as also suggested by another laboratory [39]. We identified 51 genes related to the regulation of histone functions. The Venn diagram analysis in Appendix A indicated that three histone deacetylases (*hdac1, hdac3* and *hdac8*) are shared by all tips (v-GSR), but histone acetyltransferase (*hat1*) is present exclusively in the v-GS. There are two unique genes in the v-R: *elp3* and *usp22*. Since they may play a role in axon retraction, they were selected for validation by real-time q-PCR.

### 3.6. Validation of Genes by Real-Time q-PCR, IHC and Fluorescence Labeling

Based on the results from DESeq2 and the Venn diagram analysis, we selected several pivotal genes to validate their expression with q-PCR, including five conventional MAPKs (*erk*, *jnk1/2* and *p38α/β*), two genes limited to the v-R region of the Venn diagram (*elp3* and *usp22*) and two genes differentially expressed in G tips (*map3k2*, *csnk1e,* G > S and R in Figure 3D). For conventional MAPKs, *erk* was upregulated both in G and R tips. A *jnk* isoform, *jnk2*, was downregulated in G tips, but *p38α* was upregulated (Figure 8). Consistent with the finding in Figure 3, *map3k2* and *csnk1e* were differentially expressed in G tips (G > S and R), whereas *elp3* (not shown) and *usp22* were differentially expressed in R tips (R > G and S; Figure 8). The inconsistency for the Erk gene in R tips between Figure 7 and Figure 8 is caused by the cut-off line set up for the Venn diagram. The Erk gene was removed from the R tip list because only one tip (R3) expressed the erk gene at a very high level.

To further confirm the co-expression of map3k2 and csnk1e in growing axon tips, IHC was performed using antibodies, whose qualities were checked by Western blotting (Figure 9A and Appendix A). Brain frozen sections from uninjured lampreys were incubated with antibodies against map3k2 and csnk1e (Figure 9B,C). The slides were stained by SignalStain DAB Chromogen (brown, Cell Signaling Technology) and counter-stained with hematoxylin (blue). The cytoplasms, nuclei and axono-dendritic trees of three giant neurons (M3, I1 and Mauthner neuron) were stained with the map3k2 antibody (Figure 9B). In contrast, the csnk1e antibody stained predominantly the nucleus, with only faint staining in the cytoplasm or axoplasm. As controls, frozen sections from uninjured lamprey spinal cords were double labeled with both antibodies. A low magnification image of fluorescence double-labeling is shown in Figure 9D, which covers half of the spinal cord in a horizontal section. At higher magnification (Figure 9(E1–E4)), axonal staining with both antibodies was weak compared with the cytoplasm and nuclei of adjacent cells (cells 1–4 in Figure 9(E4)). The same protocol was implemented on three types of axon tips (G, S and R; Figure 10). The colocalization of map3k2 and csnk1e was found in G tips (Figure 10G–I) and S tips (Figure 10J–L), but not in R tips (Figure 10M–O).

### 3.7. Network Analysis of Genes Related to map3k2 and Csnk1e

As a result of their apparent prominence in differentiating growing tips from static or retracting ones, we performed network analyses of the proteins related to map3k2 (Figure 11) and csnk1e (Appendix A). The map2k3 network consists of 32 nodes, of which 27 are connected and 5 are isolated. Most of these proteins are the members of the three-tiered MAPK pathway, including MAPK kinase kinase (MKKK), MAPK kinase (MKK), and MAPK. Among them, *map2k4* and *mapk8(1)* were the most upregulated genes found in G tips and S tips, respectively. Enrichment analysis of the listed 32 genes indicated that they belong to the “MAPK signaling pathway” (FDR = 1.47× 10^−12^), or more specifically, the “JNK cascade” (FDR = 3.98 × 10^−6^). A pivotal node “jun” is missing from the RNA-sequencing list because it has not been annotated in the lamprey genome. The csnk1e network consists of 24 nodes, among which 9 genes (green color) were highly upregulated in one of the three tip types. GO enrichment analysis showed that they are related to “circadian rhythms” (FDR = 7.14 × 10^−18^), or “circadian regulation of gene expression” (FDR = 1.37 × 10^−14^). Data used for constituting 2 PPI networks are provided in Appendix A.

## 4. Discussion

After axotomy, the neuron responds by turning on genetic programs that either promote axon regeneration or signal neuronal death. Which response pattern predominates may depend on environmental factors, such as the presence of growth-inhibiting molecules [40]. However, within the same animal and the same extracellular environment, regenerative ability depends in part on neuronal identity [21,22]. Studies based on ISH and IHC of neuronal perikarya in the lamprey and zebrafish have been particularly useful in identifying molecular markers that distinguish good regenerators from bad ones [23,41,42,43,44,45]. Since regenerative ability is a stochastic property, in most cases, it is not possible to know whether at the time of examination that the cell’s axon was actually regenerating, and thus, the relationship between the markers and axon regeneration was not precise. This would be particularly important because, in large animals, the axon is long and responses to injury that rely too heavily on transcription in the cell body with subsequent anterograde transport may be too slow to be effective. The combined delays involved in retrograde transport of injury signals, transcription, translation and anterograde transport of product proteins has been invoked as a reason that local translation is required at the regenerating axon tip [7]. In a previous study, we showed that, in the lamprey, where axon regeneration occurs after spinal cord TX, mRNAs, including ones coding for some cytoskeletal proteins, are present in the injured axon tips and are more abundant in the tips of actively growing axons than in static or retracting ones. EM observations demonstrated the presence of the elements of translational machinery in the axon tips, including ribosomes/polyribosomes, ribosomal protein S6 and even structures similar to rough endoplasmic reticulum [29]. In the present study, using scRNA-seq, we analyzed the gene expression profiles in the axoplasms of axon tips in different growth states (growing, static and retracting) post-TX and showed the presence of mRNAs directly related to protein synthesis and to MAPK pathways, selectively in these axon tip types. Thus, the present study does not correlate the expression of specific genes with the probability that a neuron’s axon will regenerate after injury, but with whether the axon is actually growing at the time of sampling. Among the transcripts identified are large groups of previously unknown functions or those belonging to pathways not previously implicated in the regulation of axon elongation or retraction. Their roles in axon dynamics remain to be clarified.

### 4.1. Transcriptomes Prepared from Individual Axon Tips in Three Stages of Axon Growth

Many investigations have been performed to elucidate the mechanisms of axon regeneration by identifying mRNAs in neurites. Early techniques, such as conventional PCR methods or microarray analysis, have low efficiencies and detect only dozens or hundreds of transcripts [46,47]. With the advent of new technologies, especially next-generation sequencing (RNA-seq), several thousand genes are detected from transcriptomes prepared from mouse and squid axons [14,48,49]. The combination of this new technology with the single axon tip sampling in the present study offers the possibility of deeper insights into the mechanisms underlying the maintenance of normal axonal functions and the responses to pathological stimuli, such as axotomy. The growth statuses of axon tips were determined in 4 h windows, followed immediately by micro-aspiration. This provided us an opportunity to view mRNA profiles directly related to the growth status for a given axon tip. Previously, evidence suggested that axon regeneration is discontinuous, with periods of forward growth interrupted by periods of stasis or retraction [28]. For some tips, retraction is irreversible, and the neuron is destined to die (e.g., axon tip 2 in Figure 1). The micro-aspiration of axoplasm in these retracting tips often was difficult, possibly due to leakage of their water-soluble contents, leaving only very viscous axoplasm. From the Venn analysis in Figure 7, we can see that RNA-seq identified 6437 unique gene species represented in the axoplasms of injured axon tips at 10–14 days post-TX. This is comparable to the 6118 mRNA species found in axons of mouse sensory neurons after growth for 48 h in vitro [14], although the growth status of those axons was not known. In the present study, growing tips contained transcripts for the largest number of gene species (5097), but of these, only 1632 (32%) were unique to growing tips. Similarly, of the 4105 genes represented in static tips, only 837 (20%) were specific to the static state, and of 2318 genes represented in retracting tips, only 316 (14%) were specific to retraction. Of the total of 6118 mRNA species found in axon tips, 1431 genes (23%) were common to all three groups. An additional 2221 genes (36%) were shared between G-S, G-R or S-R tips. Thus, most of the genes (60%) represented in injured axon tips were not associated uniquely with the axons’ growth statuses. However, growing tips had the largest diversity of mRNAs and the largest percentage of mRNAs unique to a specific growth state. This adds weight to the hypothesis that local protein synthesis is involved in the mechanism of axon regeneration.

### 4.2. Genes Directly Related to Axon Regeneration


(1)*map3k2 and csnk1e*: From the G > S and R list in hierarchical cluster analysis (Figure 3D), the protein encoded by *map3k2* is involved in the MAPK pathway, and the protein encoded by *csnk1e*, casein kinase 1 epsilon, is involved in circadian rhythms. Results from qPCR (Figure 8) and IHC studies (Figure 10) confirmed their presence in axon tips. The colocalization of map3k2 and csnk1e in growing tips, and their parallel expression levels in three different tips, suggests cross-talk between the MAPK pathway and circadian rhythm proteins in the regulation of axon elongation. Indeed, such cross-talk has been described previously for MAPK in the regulation of circadian rhythms [50]. Their roles in axon regeneration are discussed below.(2)Genes involved in ribosomal function: Approximately 300 of the genes found in axon tips are believed to be directly involved in RNA processing, ribosome biogenesis and protein synthesis (Figure 5).(3)G > R genes: Among the top 10 genes upregulated more in growing tips than in retracting tips (Table 2), *eif2s1a* and *eif4b* are translation initiation factors. The protein encoded by *alas2* catalyzes the first step in the heme biosynthetic pathway.(4)Genes related to histone modification: Genes, including *hadac1*, *hadac3* and *hdac8*, encode histone deacetylase isoforms 1, 3 and 8, respectively, found in all three types of tips (G, S and R) are thought to act in maintaining the stability of chromatin structures (Appendix A). The gene *hat1* encodes histone acetyltransferase-1, which modifies DNA transcription, and is found in G and S tips.(5)Genes related to RNA binding.


RNA-binding proteins (RBPs) bind to RNA in cells and participate in the formation of ribonucleoprotein complexes. RBPs contain structural motifs for binding to specific regions of RNA. They play major roles in the post-transcriptional control of RNAs, including RNA transport, splicing, polyadenylation, stabilization, localization and translation. GO enrichment analysis from DEGs in G vs. R tips (Figure 5) revealed 87 genes related to poly(A) RNA functions (Appendix A), including mRNA processing (19 genes), isopeptide bonding (30 genes), RNA splicing (17 genes), UBI conjugation (33 genes), ribosome biogenesis in eukaryotes (7 genes), etc. Genes related to mRNA transport include *casc3*, *lrpprc* and *slbp*; genes related to ribonucleoprotein granule formation include *dhx30*, *rpusd4* and *tbrg4*; and genes related to signal recognition particles include *srpra* and *srp72*.

### 4.3. Genes Related to Axon Retraction

(1)Genes found in the R > S and G list (Figure 3D): Among the genes expressed selectively in retracting axon tips, *dcaf17* encodes a protein associated with cullin 4A/damaged DNA binding protein 1 (DDB1), which is involved in protein degradation. The gene *prkcq* encodes protein kinase C theta, which is involved in T cell activation and other functions.(2)Genes found in the G < R or R > S lists: Among the top 10 genes expressed at higher levels in retracting than in growing tips (G < R, Table 2) or in retracting than in static tips (R > S, Table 4) are *cdk7*, *rxfp3*, *oprm1*, *grm7*,and *sdc2*. Proteins encoded belong to the “Gα(i) signaling events” family (STRING, FDR = 0.035). They inhibit the cAMP-dependent pathway through the inhibition of adenylate cyclase [51]. Genes *prkcq* and *sdc2* encode protein kinase C theta (above) and syndecan 2, and both participate in apoptosis of osteoblasts [52].(3)Necroptosis-related genes: The gene *usp22* (Appendix A) encodes the protein ubiquitin-specific peptidase 22, which has H4 histone acetyltransferase activity (GO terms, DAVID). The loss of USP22 expression significantly delays TBZ (TNFα/Smac-mimetic/zVAD.fmk)-induced necroptosis [53].

### 4.4. Genes Involved in Local Protein Synthesis and Mitochondrial Function

Many mRNAs coding for ribosomal proteins were present in all three types of axon tip in this genome-wide analysis. There were 130, 121 and 87 transcripts that encode ribonucleoprotein and 66, 51 and 36 transcripts that were involved in “protein biosynthesis” expressed in G, S and R tips, respectively. Genes involved in mRNA (G = 114, S = 98, R = 50), tRNA- (G = 27, S = 17, R = 7) and rRNA- (G = 38, S = 32, R = 17) processing also were present in the transcriptomes, suggesting that the maturation of mRNA, tRNA and rRNA occurs in axonal compartments. Interestingly, “mitochondrion” transcripts were the first and second significant GO terms in the G vs. R (Figure 5) and S vs. R (Figure 6) comparisons, respectively, suggesting an essential role for mitochondrial function in axon elongation. There were 327, 272 and 135 genes in G, S and R tips encoding mitochondrial proteins. As in humans, genes in lamprey mitochondria only encode 13 known proteins [54,55]. Therefore, transcripts for mitochondrial proteins found in axoplasms are mostly from nuclei. This suggests that, in the lamprey, as in mammals, there is a mechanism for importing proteins synthesized in axoplasms into the mitochondrion [56]. A total of 81 genes were upregulated in G tips significantly more than in R tips (Figure 5A). These genes coded for proteins found in the mitochondrial matrix (39 genes), inner membrane (31 genes) and outer membrane (7 genes), participating in electron transport (*ndufs1*, *cyc1*, *etfdh*, *sdha*, *uqcrc1*) or oxidoreduction reactions (*bdh1*, *ndufs1*, *acadsb*, *aldh1l2*, *alkbh1*, *etfdh*, *idh3a*, *mecr*, *sdha*, *sqor*). Cells are powered by two main energy sources: glycolysis and mitochondrial respiration. The energy for local protein synthesis in neurites is mostly from mitochondria. The synthesis of protein granules in the dendrites of rat primary hippocampal cultures was inhibited by the ATP synthase inhibitors, antimycin or oligomycin [57]. In *Xenopus laevis* embryos, approximately 35% of endosomes (a new translation platform) were found adjacent to mitochondria, and 80% of these were associated with RNA granules [58]. This idea is supported by our previous electron microscopic observation of a mitochondrion abutting a structure similar to rER in an axon tip [29].

### 4.5. MAPK Pathway in Axon Regeneration

The basic assembly of MAPK pathways is a three-component module composed of MAPK kinase kinase (MKKK), MAPK kinase (MKK) and MAPK. In mammals, there are 14 MAPKs in 7 groups. Conventional MAPKs consist of the extracellular signal-regulated kinases 1 and 2 (ERK1/2), c-Jun amino (N)-terminal kinases 1, 2 and 3 (JNK1/2/3), p38 isoforms (α, β, γ, and δ) and ERK5 [59]. The lamprey has only one erk (erk2-like) gene [3], consistent with a search of the updated assembly of the lamprey genome [60]. Of interest, we have reported that lamprey erk was activated dramatically almost immediately after spinal cord TX, and the activated form (p-erk) was rapidly transported retrogradely [3]. However, the total amount of erk protein was increased much less and only gradually, suggesting that the erk may have not been locally translated. In the present study, erk mRNA was expressed in all injured axon tips. It is not known what role erk plays locally in axon dynamics. A search of the NCBI GenBank indicates that there are two jnk (jnk1, jnk2) and two p38 (p38α, p38β) isoforms. Mammalian JNK, as well as p38, are strongly activated by various environmental stresses and inflammatory cytokines. MKK4 (MAP2K4) and MKK7 (MAP2K7) are the major protein kinases responsible for JNK activation, and MKK3 (MAP2K3) and MKK6 (MAP2K6) are the activators for p38 [59]. Most stimuli that activate JNK MAPKs also stimulate p38. Both JNK and p38 phosphorylate a large number of substrates [61,62]. Most of them are transcription factors regulating inflammatory processes, cell survival and proliferation and apoptosis. Each pathway has dual effects on cell functions, e.g., in the regulation of apoptosis. JNKs play a role in the induction of apoptosis but also have been implicated in enhancing cell survival and proliferation [63]. The opposing roles of JNKs have been attributed to the activation of different substrates depending on the specific stimulus, cell type or temporal aspects [64]. DLK-1 promotes axon regeneration in *C. elegans* through a p38/MAPK-like pathway [65,66,67]. However, overexpressed DLK-1 in Purkinje cells caused both rapid and slow degeneration or apoptosis through the JNK/MAPK pathway [68]. The same is true for Erk/MAPK signaling. Erk has been implicated as an initial signal of injury in neuronal apoptosis [69], neurite outgrowth [70] and cell proliferation [71]. The duration, magnitude and/or compartmentalization of activation determine the cellular outcome. Transient activation of Erk by growth factors results in the promotion of neuronal survival, whereas sustained activation may promote neuronal death, which can be mediated by its own pathway or through a stress kinase (p38/JNK) pathway [72]. Moreover, cross-talk between pathways can take place at many levels—MKKKs, MKKs or MAPKs [61,71]. The diversity of MAPK isoforms makes it even more complicated to assign specific cellular functions to MAPK signaling. For example, JNK1 and JNK2 played primarily opposing roles in mucosal hyperplasia and neutrophil recruitment early in otitis media [73]. For all these reasons, it is helpful that we have been able to analyze the mRNA expressions of these various genes in a specific cellular compartment (the injured axon tip) and during specific axonal activities (growth, stasis and retractions).

Based on our RNA-seq and q-PCR results, we hypothesize that, in the lamprey, three conventional MAPK pathways coordinate in regulating axon dynamics (Figure 12). The erk/MAPK signaling pathway forms the background on which different isoforms of jnk or p38 drive the axon either into regeneration or retraction. Among them, p38α mediates the process of axon elongation, whereas jnk2, together with p38β and p38γ, regulates the process of axon retraction. Since it exists predominantly in S tips, jnk1 might be involved in axon anchoring, mediated through proteins such as those encoded by *wisp2*, *col19a1* and *ptprk* (Table 3 and Table 4). This hypothesis is supported by the following facts: (1) our own unpublished data indicate that the inhibition of the Erk/MAPK pathway by U0126 promotes axon regeneration in vivo; (2) the regeneration of large-diameter axons in p38α mutant mice is reduced compared to wild type mice [74]; and (3) opposite functions were found in isoforms of JNK1 and JNK2 [73].

### 4.6. Csnk1e/Circadian Rhythm Network in Axon Regeneration

Gene csnk1e and its role in axon regeneration has been noted because of its upregulation in G tips compared to both S tips and R tips. When we performed the GO enrichment analysis, this gene was among the three (*map3k2*, *csnk1e* and *smo*) that were annotated to participate a pathway in the BIOCARTA database (Appendix A). Its increased expression was confirmed in q-PCR and IHC experiments. A PPI network was created based on the Medline database and was shown to have the properties of a circadian rhythm pathway. It has been known that mitosis is more or less likely to occur at certain times of day in mammals [75,76]. The circadian clock controls temporal hepatocyte division during liver regeneration [77]. The central module in the circadian rhythm pathway contains the CLOCK/BMAL1 complexes, which control the transcription of genes involved in the cell cycle, cell proliferation and differentiation, DNA repair and apoptosis. Their inhibitory proteins, PER and CRY, can be phosphorylated by casein kinase 1ε/δ (CK1ε/δ, csnk1e/csnk1d) and adenosine 3′5′-monophosphate (AMP) kinase (AMPK), respectively, to promote their degradation by the 26S proteasome complex through ubiquitination. Reductions in PER and CRY protein levels relieve the suppression of CLOCK/BAML1 activity, thereby permitting the establishment of a new oscillatory cycle [78]. Therefore, if G tips contained more csnk1e protein, this may promote axon regeneration by enhancing the degradation of inhibitory proteins. Additional mechanisms are possible, since it has been reported that csnk2α1, another casein kinase, phosphates the stress granule protein G3BP1 in injured axons, leading to the disassembly of the granule, the release of local mRNA and the promotion of axon regeneration [79]. However, the roles played by circadian rhythm proteins in axon regeneration remains to be clarified.

### 4.7. Genes Smo and Prkcq in Axon Regeneration

Gene *smo* is reported by the BIOCARTA database as a member of the sonic hedgehog (Shh) pathway (Appendix A). Hedgehog signaling is important in embryonic development and tumorigenesis signaling in two major ways: (A) the canonical pathway; and (B) non-canonical Shh signaling. A protein encoded by *smo*, a G-protein-coupled receptor, mediates the first pathway. The cascade is relayed by Gli family proteins (Gli1, Gli2, etc.). This pathway is inhibited by another 7-transmembrane protein Ptch1 and is released when it binds to ligand Shh. The regulation of gene transcription results from the translocation of Gli family proteins to the nucleus [80]. The stimulation of the canonical pathway by the Shh protein enhances neurite elongation in pelvic ganglia cultures [81]. The protein encoded by *prkcq* is a member of the PKC family. It is a calcium-independent and phospholipid-dependent protein kinase [82], involved in multiple pathways, including keratinocyte differentiation by BIOCARTA and NF-kappa B (NF-κB) signaling pathways, the T cell signaling pathway, etc., by KEGG. It is the only kinase that is downregulated in both G and S tips. The true function of *prkcq* in the nervous system is unclear. It may function through the activation of the NF-κB transcription factors, as occurs in the immune system [83]. NF-κB is a pleiotropic regulator in multiple neurological functions and diseases [84,85]. A role for *prkcq* in neural degeneration and regeneration has been evaluated recently [86]. Prkcq expression decreased significantly during sciatic nerve repair, which is in line with our findings that the gene is differentially downregulated in G vs. S tips. Although this phenomenon remains to be clarified, the upregulation of Prkcq activity may be involved in neuro-degradative processes.

## 5. Conclusions

Although they are far from an in-depth analysis of the roles of all kinases of the MAPK pathways in regulating axon dynamics, the results of this genome-wide gene expression analysis provide multiple targets for further studies aimed at understanding the mechanisms of post-injury axon dynamics. the selective upregulation of a subset of MAPK pathways, such as map3k2/map2k4/p38α, may be an effective strategy to improve axon regeneration. The present study provides the first overall picture of a complex signaling network involving different subsets and isoforms in axon elongation and the opposite process of axon retraction. In addition, in lamprey, there are close correlations among the distances of axon retraction, regeneration failure and delayed apoptosis [23,25,26,87].
This study suggests that genes closely associated with axon retraction (e.g., *dcaf17, prkcq*, *cdk7*, *rxfp3*, *oprm1*, *grm7* and *sdc2*) may also be involved in signaling regenerative failure and delayed cell death. The present study also suggests possible cross-talk between the MAPK and circadian rhythm pathways, which previously seemed unrelated to the mechanisms of axon regeneration. We recognize that the descriptive results obtained in this study require validation by molecular manipulations. However, the study includes a great deal of valuable data that we and others can mine to make many more discoveries about the role of local protein synthesis in axon regeneration and about the roles of specific proteins in regeneration and degeneration. We used morpholino antisense oligonucleotides to conduct knockdown studies in the lamprey, and we may be able to do so with a very small number of genes selected on the basis of the uncovered data. However, with its very long life cycle, the lamprey may not be the best animal in which to test the functions of discovered local mRNAs. Our data may point the way to experimental manipulations in species, such as mice, that are more convenient for molecular manipulation, but in which it would be very difficult to obtain the information contained in the present report.

## Figures and Tables

**Figure 1 cells-11-02320-f001:**
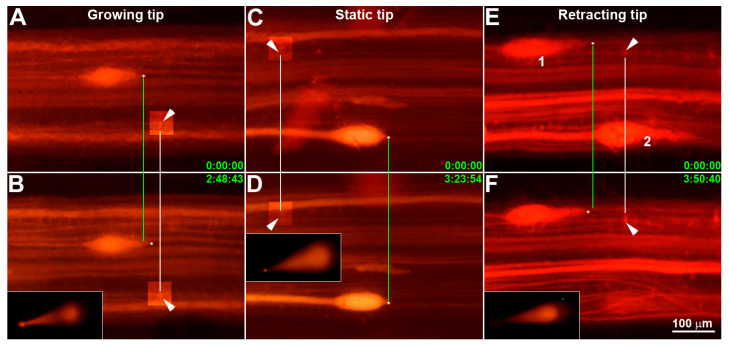
Sampling axoplasm by micro-aspiration from growing, static and retracting axon tips. (**A/B**), (**C/D**), (**E/F**), three SCs containing a growing (+21.5 µm), a static and a retracting (−13.6 µm, **1**) tip, respectively. Another tip (**2**) disappeared after 3.8 h of observation (**F**). Insets in (**B**,**D**,**F**): fluorescent images of glass micropipettes containing axoplasm following successful micro-aspiration. White lines: Alignment of fiduciary landmarks (local neuronal cell bodies). Green lines connect white dots labeling the leading edge of axon tips in the initial fluorescent images. Six green numbers indicate the times when the 1st and 2nd images were captured. The six arrowheads point to fiduciary landmarks. In the four small square regions of (**A**–**D**), brightness and contrast have been adjusted to show fiduciary landmarks.

**Figure 2 cells-11-02320-f002:**
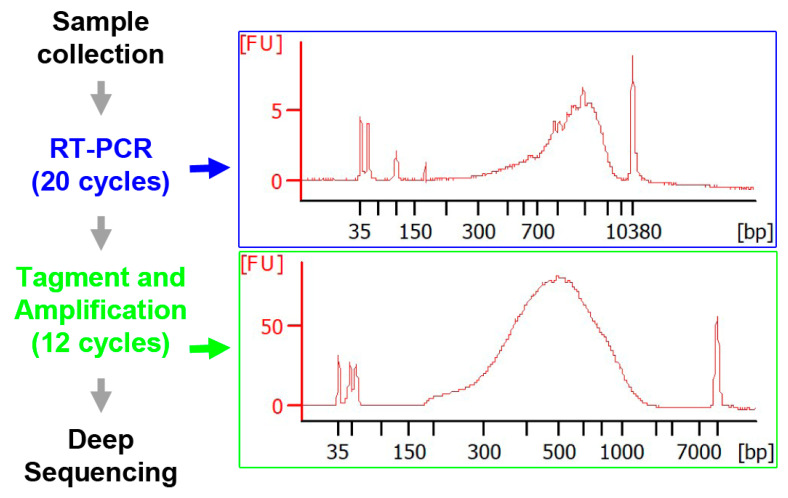
Flowchart for the construction of NGS libraries and sequencing. Axoplasms from 10 growing, 9 static and 5 retracting tips were collected, reverse transcribed and pre-amplified with a SMART-Seq Single Cell Kit (Takara Bio, San Jose, CA, USA). To construct the NGS libraries, cDNAs were tagmented and amplified with a Nextera XT DNA Library Prep Kit (Illumina, San Diego, CA, USA). DNA qualities from two reactions were checked with a 2100 Bioanalyzer (Agilent Technologies, Santa Clara, CA, USA), as shown in the two right panels. Deep sequencing was performed on an Illumina Novaseq 6000 System with paired-end reads (2 × 100 bp).

**Figure 3 cells-11-02320-f003:**
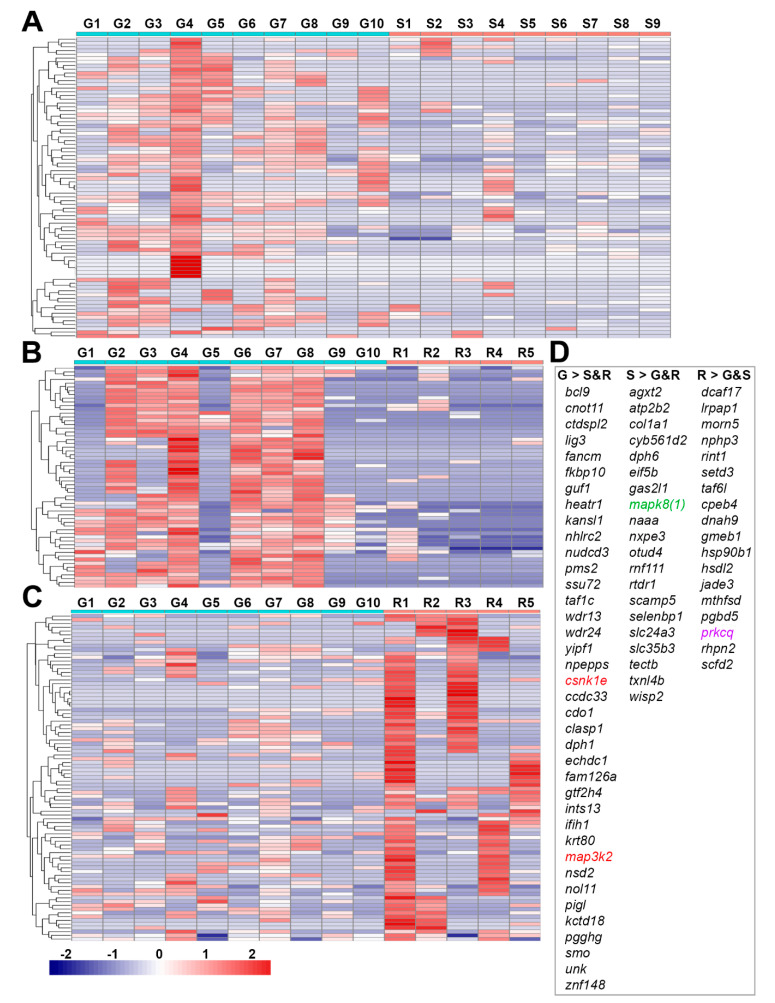
Hierarchical cluster analysis of gene expression in growing, static and retracting axon tips. Heat maps were constructed for those genes that showed significant differential expression patterns between growing (**G**), static (**S**) and retracting (**R**) axon tips: (**A**) A representative heat map cluster showing genes that were expressed significantly more in **G** than in **S** tips. (**B**) A similar heat map for genes that were expressed more in **G** tips than in **R** tips. (**C**) A similar heat map for genes that were expressed more in **R** tips than in **G** tips. The blue-to-red gradation represents levels in gene expression, from low to high. (**D**) Three gene lists identified that were *consistently* expressed more in **G** tips than in **S** and **R** tips (**G** > **S** and **R**, 38 genes), more in **S** tips than in **G** and **R** tips (**S > G** and **R**, 20 genes) or more in **R** tips than in **G** and **S** tips (**R > G** and **S**, 18 genes). Among them, *map3k2* and *csnk1e* (red font) were significantly upregulated in **G** tips more than in **S** tips or R tips and are involved either in “MAPK-signaling-cascades” or in “circadian rhythms”, respectively. The gene *mapk8(1)* (green font) in the “**S > G** and **R**” list participates in “MAPK-signaling-cascades”, and the gene *prkcq* (purple font) in the “**R > G** and **S**” list participates in the T cell receptor signaling pathway.

**Figure 4 cells-11-02320-f004:**
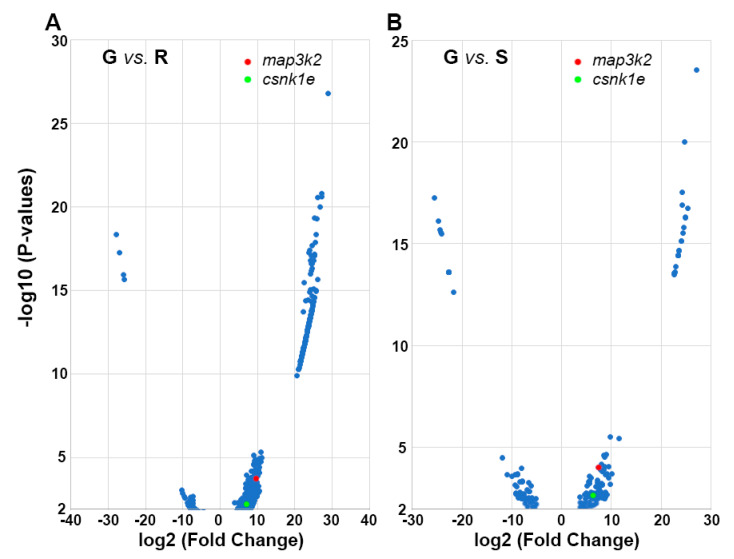
Volcano maps showing the consistently greater expressions of *map3k2* and *csnk1e* in G tips than in R tips or S tips: (**A**) Volcano plot of genes that were differentially expressed in **G** vs. **R** tips. Blue dots: 513 genes that were comparatively upregulated (right side) or downregulated (left side) in **G** tips compared with **R** tips (*p* values < 0.01). Red and Green dots represent the genes *map3k2* and *csnk1e*, respectively. (**B**) Volcano plot of 151 genes that were differentially expressed in **G** tips vs. S tips (*p* < 0.01).

**Figure 5 cells-11-02320-f005:**
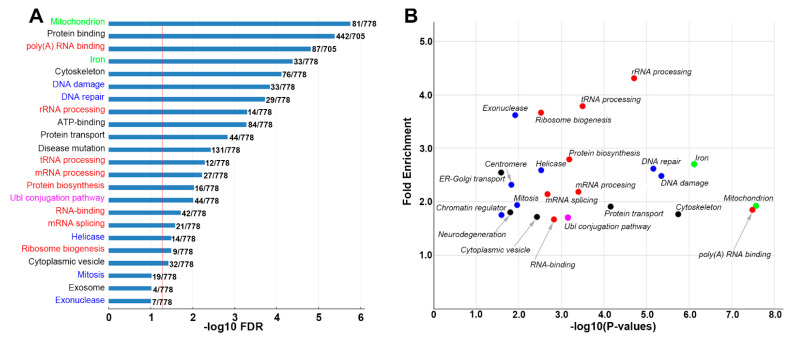
GO enrichment analysis of DEGs in G vs. R tips. A list of genes that were differentially expressed between growing tips and retracting tips (1488 genes) were analyzed by DAVID enrichment analysis (v. 6.8): (**A**) GO terms for genes were expressed more highly in **G** tips than **R** tips (FC > 1.5, FDR < 0.1). The horizontal axis represents the −log_10_(FDR) of the significant GO terms. The x-intercept of the vertical red line indicates the standard cutoff value, where −log_10_FDR = 1.301 (FDR = 0.05). Numbers beside the bars indicate the list hit numbers vs. list total numbers. Red terms are genes directly involved in protein synthesis; Green terms are for mitochondrial function; Blue terms are for the modification of DNA or chromosomes; the Pink term is for the ubiquitin conjugation pathway; and the Black terms are for other cell functions. (**B**) Scatter plot for genes expressed more highly in G tips than R tips (*p* < 0.05, FC > 1.5). Color dots represent categories of GO terms similar to those in (**A**).

**Figure 6 cells-11-02320-f006:**
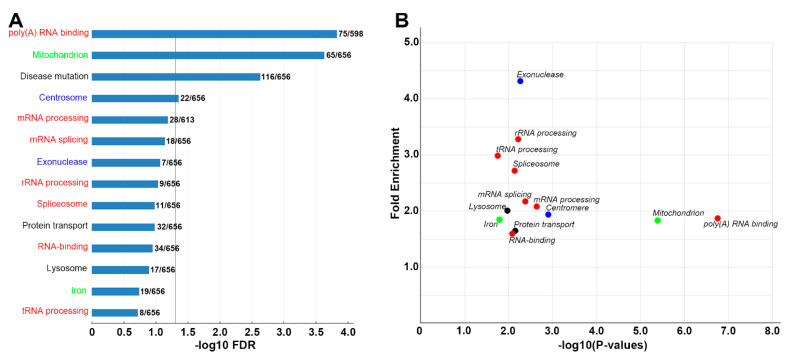
GO enrichment analysis of DEGs in S vs. R tips. A list of genes that were differentially expressed between static tips and retracting tips (1067 genes) and that were identified and analyzed by DAVID enrichment analysis: (**A**) GO terms for genes expressed more highly in S than in R tips (FC > 1.5, FDR < 0.2). The x-intercept of the vertical red line indicates the standard cutoff value, in which −log_10_FDR = 1.301 (FDR = 0.05). Red terms are genes directly involved in protein synthesis; Green terms are mitochondrial function genes; Blue terms are genes involved in the modification of DNA or chromosomes; and Black terms are genes involved in other cell functions. (**B**) Scatter plot for genes expressed more highly in **S** tips than **R** tips (*p* < 0.05, FC > 1.5). Color dots represent categories of GO terms, similar to those in (**A**).

**Figure 7 cells-11-02320-f007:**
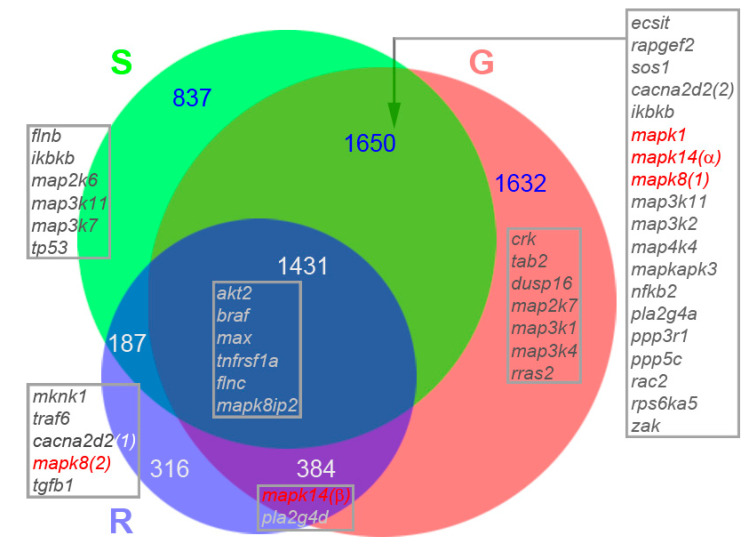
Multiple transcripts for proteins of MAPK signaling pathways were identified in growing, static and retracting axon tips. Genes obtained by RNA-seq from axoplasms of growing (G; 5097 genes), static (S; 4105 genes) and retracting (R; 2318 genes) tips are mapped in a Venn diagram. Genes participating in MAPK pathways in each sub-group were identified (DAVID) and listed inside the gray frames. Red text is used for the conventional MAPK genes: *mapk1* = *erk*, *mapk14* = *p38* and *mapk8* = *jnk*. The Venn analysis suggests that the Erk/, Jnk/ and p38/MAPK pathways are heavily involved in axon regeneration dynamics.

**Figure 8 cells-11-02320-f008:**
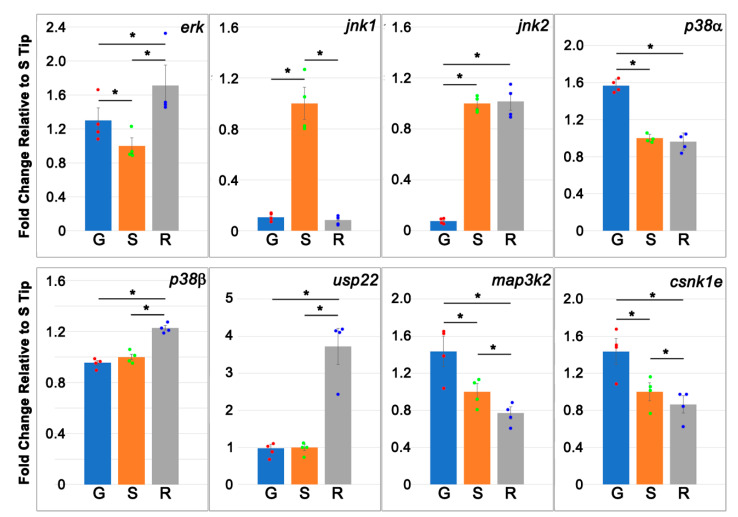
q-PCR validation of mRNA expression for key genes found in growing, static and retracting tips, including conventional MAPK genes. Blue, orange and gray bars represent gene expression levels in growing (G), static (S) and retracting (0R) tips, respectively (means ± SEM, *n* = 4). R tips contain higher levels of *erk, jnk2, p38β and usp22* genes than do G tips, whereas G tips have more *p38α*, *map3k2* and *csnk1e* than do R tips. * *p* < 0.05 by one-way ANOVA, followed by Tukey’s test.

**Figure 9 cells-11-02320-f009:**
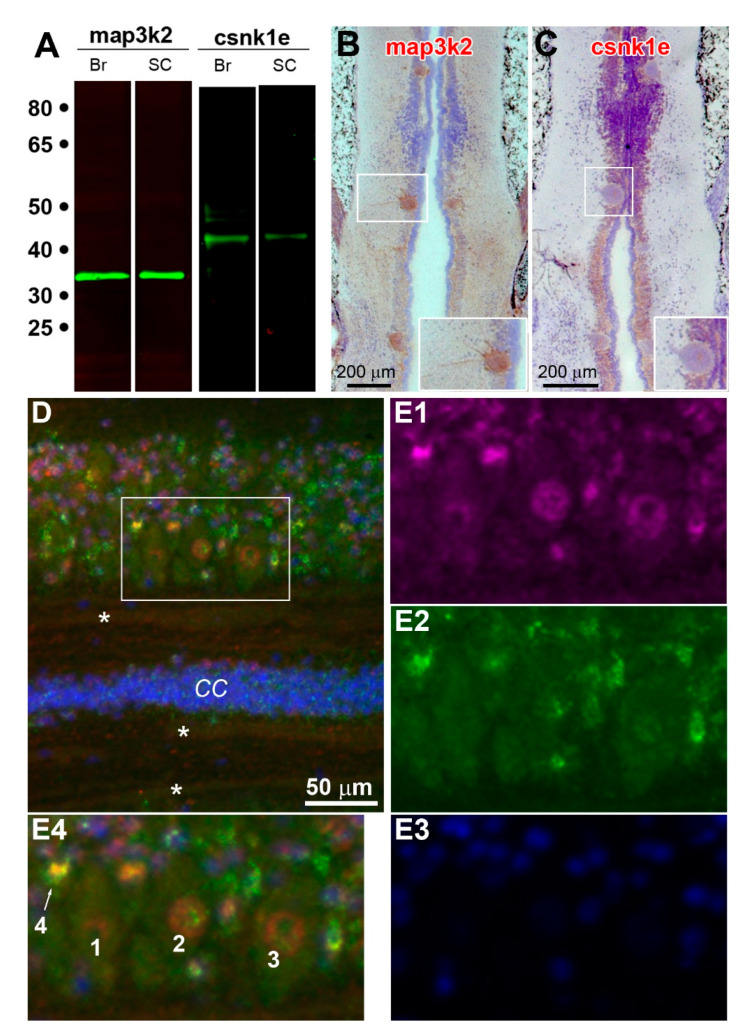
Cellular localization of map3k2 and csnk1e in frozen sections of naïve lamprey brains and spinal cords. (**A**) Western blot of proteins prepared from lamprey brains (**Br**) or spinal cords (**SC**), probed with antibodies against map3k2 and csnk1e. A single band at approximately 34 and 44 KDa was labeled with antibodies to map3k2 and csnk1e, respectively, in either brains or spinal cords. (**B**,**C**) Chromogenic IHC of coronal brain sections stained for map3k2 and csnk1e, respectively. Rostral is up. The two insets at the bottom right of (**B**,**C**) are enlarged images of the I_1_ neurons framed by the respective white boxes. Positive immunostaining is in brown; counterstaining by hematoxylin is in light blue. (**D**) Spinal cord coronal sections immuno-fluorescently triple stained for map3k2 (red), csnk1e (green) and DAPI (blue). The three asterisks (*) are inside faintly stained uninjured axons. *CC*: central canal. (**E1**–**E3**) are monochromatic enlarged micrographs from the framed region in (**D**). (**E4**) is (**E1**–**E3**) superimposed, showing local neurons (**1**–**3**) and glial cells (**4**).

**Figure 10 cells-11-02320-f010:**
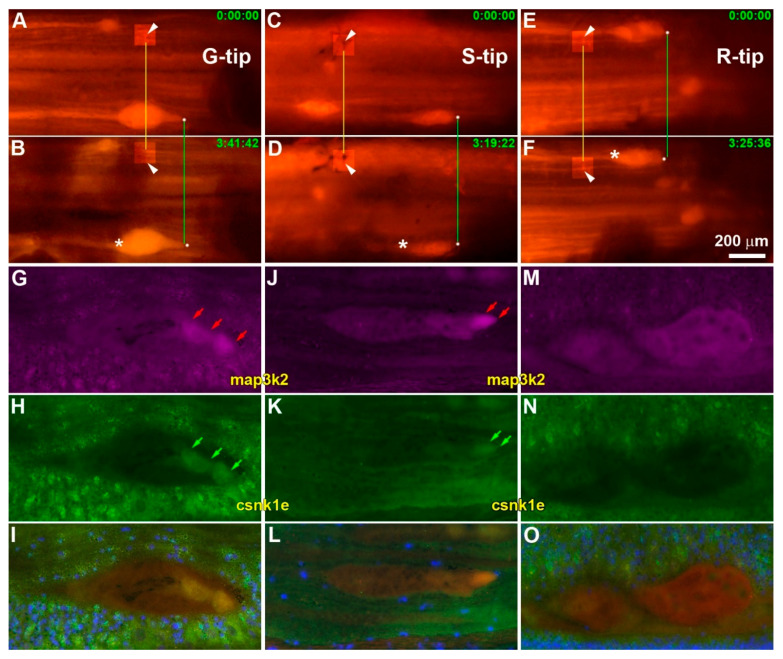
Colocalization of map3k2 and csnk1e in growing axon tips. (**A/B**), (**C/D**) and (**E**/**F**): Three spinal cords containing either a growing tip (**G-tip**; **A/B**), a static tip (**S-tip**; **C/D**) or a retracting tip (**R-tip**; **E/F**), respectively (labeled with white asterisks). Their growth statuses were determined by fluorescence photomicrography, with 3–4 h intervals between captures, as indicated by the times listed in green at the top right of each frame. Spinal cords were frozen immediately and sectioned. (**G/H**), (**J/K**), and (**M/N**): Immunofluorescence for map3k2 (red) and csnk1e (green). Axon tips from Growing or Static axons were stained by both map3k2 (**G**,**J**, red arrows) and csnk1e (**H**,**K**, green arrows). Retracting tips showed only background staining (**M**,**N**). (**I**,**L**,**O**) are the superimposed images of (**G**/**H**), (**J**/**K**) and (**M**/**N**), respectively. The blue color in (**I**,**L**,**O**) is DAPI stain. Thus, growing tips, and to a lesser degree static tips, expressed both map3k2 and csnk1e, whereas retracting tips showed only faint, diffuse map3k2 staining. Six white arrowheads point to fiduciary landmarks. Brightness and contrast in six small square regions in (**A**–**D**) are adjusted to show fiduciary landmarks.

**Figure 11 cells-11-02320-f011:**
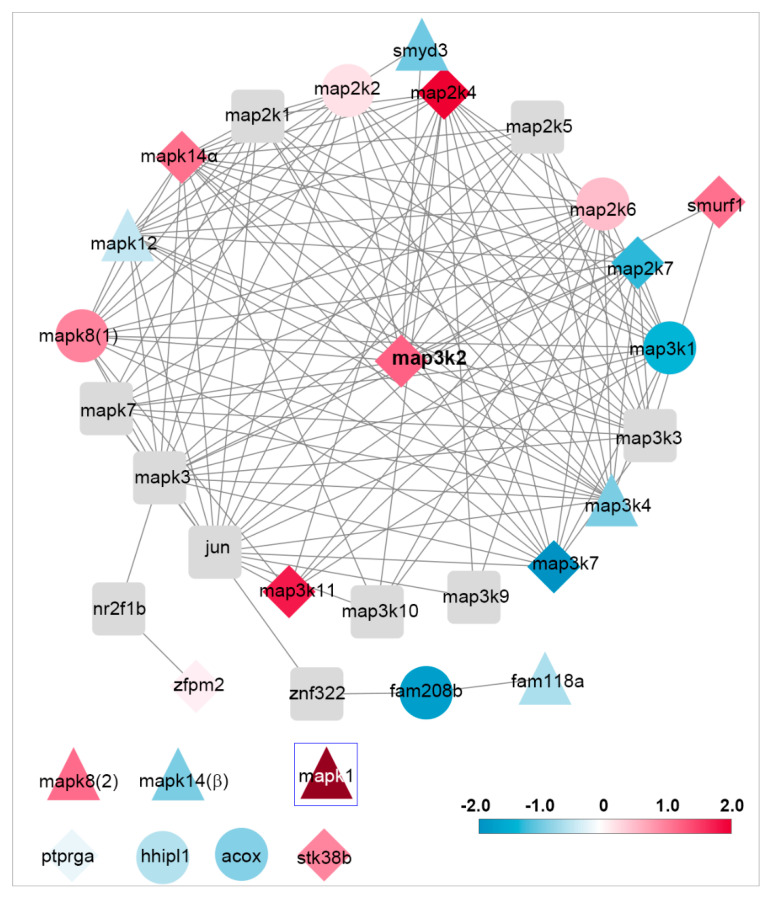
Network analysis of gene expression related to “map3k2”. Blue to red graduation represents gene expressions from low to high. Diamond, round or triangle nodes represent genes that are predominant in growing, static or retracting tips, respectively. Rectangular gray nodes represent genes that are important but not present in the sequencing lists. Six nodes (bottom left) are isolated because of their indirect linkage to the network (ptprga, hhipl1, acox, stk38b), or possibly different connections from their isoforms [mapk8(2), mapk14(β)]. Node mapk1 (erk) is referenced for its particularly high expression level (2.9) in retracting tips. Enrichment analysis indicated that nodes in this network are related to the “MAPK signaling pathway” or to the “JNK cascade”.

**Figure 12 cells-11-02320-f012:**
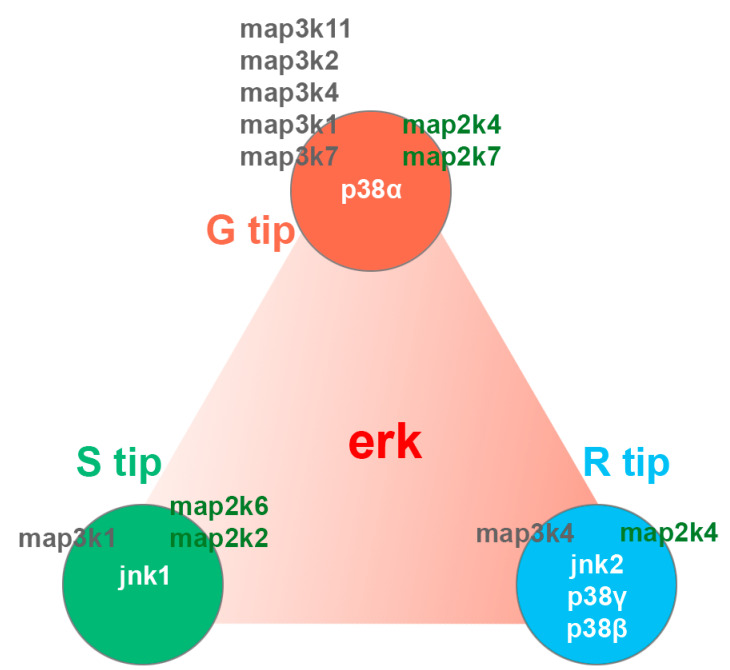
Three conventional MAPK functions in the regulation of axon regeneration coordinately and differentially. The central triangle with the red gradient indicates the platform formed by the levels of erk expression in the three types of axon tips: highest in retracting tips (100%), followed by growing tips (56%) and static tips (16%). Five isoforms (jnk1, jnk2, p38α, p38β and p38γ) of two other conventional MAPKs (jnk/p38) are differentially expressed in **G**, **S** or **R** tips. The highest expression levels were for jnk2 in **R tips** (100%) and for p38α in **G** tips (97%). This was followed by jnk1 in **S** tips (80%), p38γ in R tips (14%) and p38β in **R** tips (8%). In R tips, expression levels for erk were 8.7-fold higher than for jnk2. MAPKKK and MAPKK are listed on the left and right sides of the three circles, respectively, based on the type of tip in which they were most highly expressed (**G**, **S** or **R**), listed in order of expression levels, from highest (top) to lowest (bottom).

**Table 1 cells-11-02320-t001:** Real-time PCR primers used in the study.

Gene	GenBank Accession No.	Forward Primer 5′–3′	Reverse Primer 5′–3′
mapk1 (erk)	XR_004402373	GAGGGCGCCTACGGCATGGT	GCTGGTCGAGATAATGCTTC
mapk8_1_ (jnk_1_)	XM_032949959	GCATCTACATTCAGCTGGCA	TGTGCTAGTGCCTGATCGAC
mapk8_2_ (jnk_2_)	XM_032969277	TCACTTCTTCACCATGAGCG	GCAGTCAGACTTCACCACGA
mapk14_1_ (p38_1_)	XM_032969733	GTAATTGGGCTGCTGGATGT	ATCAGCTTCAGCTGGTCGA
mapk14_2_ (p38_2_)	XM_032954129	GACGTGAAGGAAGGCTTCTG	CAAATTGCTGGGTTTCAGGT
usp22	XM_032970308	CATGCATAGCGGAGGAAAAT	TTACACTTGCGTTCACGGAC
map3k2	XM_032951177	CTGAAAGGAACTCGCCTACG	TATTTGACAAACAGCGCCAC
csnk1e	XM_032967217	CCTACCGCGAGAACAAAAAC	GGCTGGTGTGTGGGAGAGAG

**Table 2 cells-11-02320-t002:** Top 10 upregulated and downregulated genes in G tips in comparison with R tips.

Symbol	Description	Fold Change (log2FC)	FDR	GO-Terms
**G > R**				
*alas2*	5′-aminolevulinate synthase 2	27.2	2.05 × 10^−18^	M., porphyrin biosynthesis
*gpatch11*	G-patch domain containing 11	27.2	2.05 × 10^−18^	chromosome, protein binding
*eif2s1a*	eukaryotic translation initiation factor 2 subunit alpha	26.8	6.05 × 10^−18^	C., protein synthesis
*dnaja4*	Dnaj heat shock protein family (hsp40) member A4	26.1	2.05 × 10^−18^	C., Hsp70 protein binding
*eif4b (1 of 2)*	eukaryotic translation initiation factor 4b	25.8	2.03 × 10^−17^	C., protein synthesis
*actn4 (2 of 2)*	actinin alpha 4	25.7	1.42 × 10^−16^	N./C., F-actin cross-link
*pigq*	phosphatidylinositol glycan anchor biosynthesis class Q	25.6	9.83 × 10^−14^	P., PIGQ activity
*leo1*	LEO1 homolog, Paf1/RNA polymerase II complex component	25.6	9.36 × 10^−14^	N./C., RNA transcription
*sdha (2 of 2)*	succinate dehydrogenase complex flavoprotein subunit a	25.6	4.03 × 10^−16^	M., electron transfer
*mrpl12*	mitochondrial ribosomal protein L12	25.3	2.16 × 10^−13^	M., RNA binding
**G < R**				
*dcaf17*	DDB1- and CUL4-associated factor 17	−27.9	1.42 × 10^−16^	N./C., protein binding
*hsdl2*	hydroxysteroid dehydrogenase-like 2	−25.9	1.40 × 10^−14^	M., oxidoreductase activity
*cdk7*	cyclin-dependent kinase 7	−25.7	2.47 × 10^−14^	N./C., cell cycle, RNA transcription
*oprm1 (2 of 3)*	opioid receptor mu 1	−10.4	1.05 × 10^−2^	P./ER, receptor for endogenous opioids
*mul1a*	mitochondrial ubiquitin ligase activator of nfkb 1-A	−10.0	1.48 × 10^−2^	M., protein ubiquitination
*scfd2*	sec1 family domain containing 2	−9.6	2.06 × 10^−2^	P., protein transport
*btk*	Bruton tyrosine kinase	−9.4	2.33 × 10^−2^	P./C./N., B lymphocyte signaling
*prkcq*	protein kinase C theta	−8.7	3.53 × 10^−2^	P./C., T cell differentiation
*rxfp3* *(1 of 2)*	relaxin family peptide receptor 3	−8.5	4.58 × 10^−2^	P., inhibit cAMP accumulation
*gpank1*	G-patch domain and ankyrin repeats 1	−8.4	4.87 × 10^−2^	N./C., nucleic acid binding

P. = plasma membrane, C. = cytosol, N. = nucleus, ER = endoplasmic reticulum, M. = mitochondria.

**Table 3 cells-11-02320-t003:** Top 10 upregulated and downregulated genes in G tips in comparison with S tips.

Symbol	Description	Fold Change (log2FC)	FDR	GO-Terms
**G > S**				
*mb21d2a*	Mab-21 domain containing 2	27.1	2.29 × 10^−20^	C., cadherin binding
*slc9a3*	solute carrier family 9 member A3	25.3	2.36 × 10^−14^	P., solute:proton antiporter activity
*stk19*	eukaryotic translation initiation factor 2 subunit alpha	24.9	5.31 × 10^−14^	N., N-ras signaling
*kcna4 (1 of 2)*	potassium voltage-gated channel subfamily A member 4	24.9	5.31 × 10^−14^	P., transmembrane K^+^ transport
*krt80 (11 of 11)*	keratin 80	24.7	3.90 × 10^−17^	C., protein binding
*dph1*	diphthamide biosynthesis 1	24.5	1.25 × 10^−13^	N./C., histidine modification
*smchd1*	structural maintenance of chromosomes flexible hinge domain containing 1	24.3	1.89 × 10^−13^	N., structural maintenance of chromosomes
*rtca*	RNA 3′-terminal phosphate cyclase	24.3	2.10 × 10^−14^	N., RNA processing
*tmem167b*	transmembrane protein 167B	24.1	8.30 × 10^−15^	Golgi, secretory pathway
*inhbb*	inhibin subunit beta B	24.0	3.85 × 10^−13^	N., growth factor activity
**G < S**				
*ldlrad1*	low density lipoprotein receptor class A domain containing 1	−25.6	1.17 × 10^−14^	P./Ex., protein binding
*selenbp1*	selenium binding protein 1	−24.7	7.14 × 10^−14^	N., intra-Golgi protein transport
*f2r*	coagulation factor II thrombin receptor	−24.2	1.95 × 10^−13^	P./Ex./En, phosphoinositide hydrolysis
*atp2b2 (2 of 3)*	ATPase plasma membrane Ca^2+^ transporting 2	−22.7	5.84 × 10^−12^	P./Ex., intracellular Ca^2+^ levels
*tectb*	tectorin Beta	−22.7	5.84 × 10^−12^	Ex./extracellular matrix
*gskip*	GSK3B interacting protein	−22.7	5.84 × 10^−12^	N., anchoring for GSK3B and PKA
*cyb561d1*	cytochrome B561 family member D1	−22.7	5.84 × 10^−12^	P., heme binding
*wisp2*	cellular communication network factor 5	−22.7	5.84 × 10^−12^	N./Ex., insulin-like GF binding
*ptrh2*	peptidyl-tRNA hydrolase 2	−22.7	5.84 × 10^−12^	M., hydrolase activity
*scamp5*	secretory carrier membrane protein 5	−11.9	6.33 × 10^−3^	P,/En./Golgi, exocytosis

P. = plasma membrane, C. = cytosol, N. = nucleus, Ex. = extracellular, En. = endosome.

**Table 4 cells-11-02320-t004:** Top 10 upregulated and downregulated genes in R tips in comparison with S tips.

Symbol	Description	Fold Change (log2FC)	FDR	GO-Terms
**R > S**				
*ptpra*	protein tyrosine phosphatase receptor type A	26.9	2.42 × 10^−15^	P./Ex., focal adhesion
*dnah9*	dynein axonemal heavy chain 9	26.8	2.43 × 10^−15^	CS., force-producing
*grm7*	glutamate metabotropic receptor 7	26.3	6.54 × 10^−15^	P., axon movement
*slc5a3a*	solute carrier family 5 member 3a	26.1	1.08 × 10^−14^	P., inositol transport
*morn5*	MORN repeat containing 5	25.5	3.64 × 10^−14^	CS., protein binding
*sdc2*	syndecan 2	25.3	6.75 × 10^−14^	CS., protein binding
*prkcq*	protein kinase C theta	14.2	8.18 × 10^−5^	P./CS., protein kinase C activity
*txnl4b*	Thioredoxin-like 4B	10.3	1.34 × 10^−2^	N./C., pre-mRNA splicing
*fam160b1*	FHF complex subunit hook interacting protein 2A	10.2	1.42 × 10^−2^	N./C., vesicle trafficking and/or fusion
*prrc1*	proline rich coiled-coil 1	9.9	1.88 × 10^−2^	Golgi, NA
**R < S**				
*gmppab*	GDP-mannose pyrophosphorylase A	−26.5	1.13 × 10^−14^	Ex./N., inhibition of GMPPB
*efcab6*	EF-hand calcium binding domain 6	−25.6	8.32 × 10^−14^	N./M., regulates the androgen receptor
*col19a1*	collagen type XIX alpha 1 chain	−25.3	1.29 × 10^−13^	Ex./En., cross-bridge between fibrils
*raf1a*	raf-1 proto-oncogene, serine/threonine kinase	−25.1	2.34 × 10^−13^	P./M./N./C., MAPK/ERK ptw
*gclc (1 of 2)*	glutamate-cysteine ligase catalytic subunit	−25.0	2.34 × 10^−13^	C./N., glutamate-cysteine ligase activity
*siae (1 of 3)*	sialic acid acetylesterase	−25.0	2.34 × 10^−13^	Ex./Lys, acetyl ester removal
*wdr48b*	WD repeat domain 48	−25.0	1.57 × 10^−15^	C./N., DNA repair
*trak1*	trafficking kinesin protein 1	−25.0	2.73 × 10^−13^	M./En., En.-to-Lys. trafficking
*wdr37*	WD repeat domain 37	−24.9	3.16 × 10^−13^	N./CS., NA
*ptprk*	protein tyrosine phosphatase receptor type K	−24.8	3.30 × 10^−13^	P., cell contact and adhesion

P. = plasma membrane, C. = cytosol, N. = nucleus, M. = mitochondria, Ex. = extracellular, En. = endosome, GMPPB = GDP-Mannose Pyrophosphorylase B, Lys. = lysosome, NA = not available, CS. = cytoskeleton.

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
