# Peer review of "Transcriptomes of Injured Lamprey Axon Tips: Single-Cell RNA-Seq Suggests Differential Involvement of MAPK Signaling Pathways in Axon Retraction and Regeneration after Spinal Cord Injury"

_cells, 2022, doi:10.3390/cells11152320_

Round 1

Reviewer 1 Report

This manuscript by Jin et al., provides the first comprehensive report of transcripts found in axon tips after injury. This was made uniquely possible by taking advantage of the large size of lamprey axons. The authors previously showed that they could identify growing, static, or retracting axons. Here, they develop the methods to aspirate the contents of the growing, static, or retracting axon tips, generate cDNA libraries, and perform RNA-Seq on the resulting material, comparing both the composition and levels of individual transcripts. Simply put, this is nothing short of heroic and has the potential to be a tremendous resource for the scientific community. The findings suggest particular roles for the MAPK pathway, which is known to be involved in axon regeneration across species. Additional roles for csnk1e, transcription, and mitochondria are suggested. However, there are still a few organizational issues and clarifications that need addressing in order to make the study more digestible for readers. 

1.     Understandably, the RNA-Seq generates a rather large and complex dataset that is challenging to distill into a single study. In the current presentation, the map3k2 and csnk1e emerge as “genes of interest” in the heatmap data (Fig. 3) and volcano plot (Fig. 4). However, it is not clear how these genes were selected out of the larger dataset of DE genes, which includes clasp-1 (MT protein), protein phosphatases, and other signaling genes that are known to be involved in regeneration. Moreover, mapk2k2 and csnk1e are far from being the most robust DE genes in terms of fold change (Fig. 4). How the authors arrived at studying these genes needs to be clarified better.

2.     Moreover, the manuscript would read better if the truly unbiased data were shown first (Table and GO enrichment analysis), working later towards a rationale for focusing on the mapk signaling pathway with accompanying q-PCR and imaging experiments. 

3.     In general, the focus on csnk1e was not well justified throughout the manuscript and therefore needs to be clarified.

4.     Conversely, why the robust GO enrichment of mitochondrial, RNA binding and cytoskeletal genes was not discussed in more detail is unclear (Fig. 5 and 6).

5.     In Figure 1 and Figure 10, it is difficult to see the fiduciary landmarks and the white dots, and thus hard to see how the growing, static, and retracting axon tips were assigned. Could the white dots be larger, and is there a way to make the fiduciary points more obvious?

6.     It’s not clear how the lists in Figure 3D were generated. Please clarify. 

7.     The primary data should be made available, including the DE gene lists.

Author Response

Major comments:

This manuscript by Jin et al., provides the first comprehensive report of transcripts found in axon tips after injury. This was made uniquely possible by taking advantage of the large size of lamprey axons. The authors previously showed that they could identify growing, static, or retracting axons. Here, they develop the methods to aspirate the contents of the growing, static, or retracting axon tips, generate cDNA libraries, and perform RNA-Seq on the resulting material, comparing both the composition and levels of individual transcripts. Simply put, this is nothing short of heroic and has the potential to be a tremendous resource for the scientific community. The findings suggest particular roles for the MAPK pathway, which is known to be involved in axon regeneration across species. Additional roles for csnk1e, transcription, and mitochondria are suggested. However, there are still a few organizational issues and clarifications that need addressing in order to make the study more digestible for readers

We appreciate reviewer’s favorable comments.

Minor comments:

Q1A: …… the map3k2 and csnk1e emerge as “genes of interest” in the heatmap data (Fig. 3) and volcano plot (Fig. 4). However, it is not clear how these genes were selected out of the larger dataset of DE genes, which includes clasp-1 (MT protein), protein phosphatases, and other signaling genes that are known to be involved in regeneration.

We have explained it in manuscript under subsection “DEG identification and DAVID enrichment analysis.”

Briefly we select interesting gene based on 1. Whether or not it is gene related to signaling pathways and selected by pathway database in DAVID enrichment platform; 2. Whether or not it a kinase, or kinases are preferred; and 3 Whether or not it is upregulated in G-tips since we are more interested in regeneration related genes than others.

We tested all 3 lists of genes (38, 20, 18) for pathway involvement in DAVID enrichment platform by three default pathway databases: BIDD, BIOCARTA, and KEGG. BIDD identified only one MAPK pathway related to map3k2 and mapk8; BIOCARTA generated 4 pathways, related to map3k2/mapk8 (MAPK pathway), csnk1e (circadian rhythm), smo (sonic hedgehog pathway), and prkcq (keratinocyte differentiation). KEGG presented more than 40 pathway that related to 26 genes, including that mentioned above. For the limitation of manuscript we pay attention to 4 pathways presented by BIOCARTA. Then we compared genes map3k2/mapk8, csnk1e, smo, and prkcq and found they are all kinases except smo (smoothened, frizzled class receptor), as we know many kinases plays pivotal role in multiple pathways. Inclination to find axon regeneration related genes leads us to select map3k2, and csnk1e genes for more extensive study in this research.

Q1B: Moreover, mapk2k2 and csnk1e are far from being the most robust DE genes in terms of fold change (Fig. 4). How the authors arrived at studying these genes needs to be clarified better.

See above and below. Theoretically, signaling pathway genes would be expressed at relatively low levels, so expression levels are not our most important criteria in selecting genes to focus on.

Q2: Moreover, the manuscript would read better if the truly unbiased data were shown first (Table and GO enrichment analysis), working later towards a rationale for focusing on the mapk signaling pathway with accompanying q-PCR and imaging experiments.

There are three unbiased selections in this research:

  1. A. Selecting 10, 9, 5 transcriptomes from a total 33 original samples. We have explained it in sections Methods and Material/Determination of axon tip growth status; Results/Determination of growth status…; and Results/RNA-seq, alignment and assemblies. Two supplemental data support for this unbiased selection. Supplemental data (Figure S1, Table S1) were provided to support this selection.
  2. B. Selecting 2 genes/pathways (mapk-pathway and csnk1e/circadian rhythm) out of hundreds to thousands of genes/hundreds pathways

 See answers above (Q1A)

  1. C. Selecting 5, 5, and 5 samples from 10, 9, 5 samples for Venn diagram analysis.

We select 5 G-tips, 5 S-tips transcriptomes based on their expression count correlations of each gene. A table (Table S6) was provided to support this unbiased selection.

Q3: In general, the focus on csnk1e was not well justified throughout the manuscript and therefore needs to be clarified.

See answers above (Q1A). In addition We have added a paragraph “Csnk1e/circadian rhythm network in axon regeneration” in Discussion to clarify the functions of circadian rhythm in tissue regeneration, and to indicate why it is important to extend this to axon regeneration, which is a very different matter.

Q4: Conversely, why the robust GO enrichment of mitochondrial, RNA binding and cytoskeletal genes was not discussed in more detail is unclear (Fig. 5 and 6).

As with csnk1e/circadian rhythm pathway genes, we added a section in Discussion on genes involved in RNA binding. The previous version already had a section discussing mitochondrial genes, but now we expand this under the subtitle “Genes involved in local protein synthesis and mitochondrial function”. We have provided detailed cytoskeletal genes, RNA binding genes, and mitochondrial genes, etc. in Table S5.    

Q5: In Figure 1 and Figure 10, it is difficult to see the fiduciary landmarks and the white dots, and thus hard to see how the growing, static, and retracting axon tips were assigned. Could the white dots be larger, and is there a way to make the fiduciary points more obvious?

We have adjusted the brightness/contrast and added pairs of arrowheads to better show the fiduciary landmarks, and increased white dot size.

Q6: It’s not clear how the lists in Figure 3D were generated. Please clarify.

We added an explanation under subsection “DEG identification and DAVID enrichment analysis”.

Q7: The primary data should be made available, including the DE gene lists.

These data now are provided in the 14 new items of Supplemental Data, including the entire list of genes found in all the axon tips. 

Reviewer 2 Report

This is a submittal from the laboratory most responsible for our understanding of spinal axon regeneration in the sea lamprey, and reflects the lab’s mastery of important techniques. In this manuscript, the authors describe an RNA-Seq analysis of gene expression in the tips of large lamprey spinal axons (large axons, not lampreys) in the subacute period following spinal cord transection. Some of these axons are destined to regenerate past the lesion site, while others will not. The unique contribution of this project is that, although the ultimate fate of these individual axon tips cannot be known, the authors could examine separately the genes expressed in axon tips that are advancing, retracting, or remaining static over a 4-hour period at a time (roughly 2 weeks after injury) immediately preceding RNA sampling from individual tips. This means that gene expression can be correlated with acute growth state in severed axons in vivo, which to my knowledge has not previously been accomplished.

The authors analyze their gene expression data to determine differential expression of genes in each growth state compared to the other two, and use Gene Ontology analysis to suggest subcellular compartments and/or signaling pathways that appear relatively overrepresented in the three different groups. The gene expression data are well described and interesting, and the analysis suggests some hypotheses that could be tested in future work. There are, however, some issues with methodology and analysis that, in my opinion, should be addressed prior to acceptance for publication. Some of these are enumerated below.

1)    The authors need to provide the full RNASeq data sets as supplementary files; these data sets were not provided or not available for review. The value of this work will reside largely in the availability of these data according to FAIR data principles.

2)    The authors do not describe the level(s) of the spinal cord from which the axon tips were collected. Were all tips collected from the same A-P region of the cord? Presumably this was proximal to the transection site, since it takes many weeks for axons to reach the lesion, but how far from the lesion site? Please provide this information.

3)    The impact of the work depends on the micrometer-level accuracy of measurements over the 4-hour sampling window. Therefore it would be helpful to have at least one example in which the “fiduciary marks” used for normalization can be seen in images from the two times investigated. 

4)    An FDR of 0.2 is not a common cutoff; the authors explain at one point that this was chosen so they could identify “proteins related to epigenetic modifications or…involved in signaling pathways”, but how a high FDR leads to this identification is not immediately obvious to me. Can this be clarified? In other places, FDRs of 0.05 or 0.1 are used; these choices should also be rationalized.

5)    The data are said to be expressed as mean ± SD. If this is true, data like those in Figure 8 are unusual (p38beta is particularly striking). Can the authors confirm that SD was used? If so, I recommend that the authors instead plot each data point together with means and SEM or CV.

6)    The authors should consider providing a clustering map for the 33 transcriptomes to show how the individual transcriptomes segregate (or not) with their groups. From Figure 3 it appears that this would not be completely consistent. This point relates to the PCA analysis, which indicates that the 3 groups have highly similar (indistinguishable) patterns of gene expression in the absence of two major outliers. 

7)    Selection of the 5 “most related transcriptomes” from each group was done in an attempt to reduce sample bias, but this selection has its own problems, since it reduces variability without objective criteria for deciding which tips are representative. The authors then highlight a number of genes whose functions are partially understood (from GO terms) to make general suggestions about protein function in axon tips, which is an exercise that ought to be performed with caution. Why do the authors think that genes associated with TCR signaling would be upregulated in the R axon tips, for example? What is the importance of a putative circadian rhythm-related network in axon tips? GO analysis in Figure S1 suggests that differences between growing and static tips are mainly associated with nuclei. How are these findings interpreted? Ity would be helpful to know the authors’ views here.

8)    On the other hand, there is already abundant evidence for the roles of ERKs, JNKs and p38s in axon growth and regeneration; to say that the Venn diagrams provided here suggest this idea seems to ignore a great deal of earlier work. This relates to an issue with the analysis in general; genes that authors choose to study are called “pivotal” while others with similar expression patterns are ignored. This approach seems to miss the opportunity provided by an unbiased screen of gene expression. It is recognized that there are choices that need to be made, but there should be some discussion of the pluses and minuses of this approach.

9)    In Figure S4, is there more than 1 grey rectangular node? If not, the legend should simply refer to spata32 in this context. Since spata32 protein is associated with microtubules and spermatogenesis, its presence at the center of a network supposedly related to circadian rhythms (and csnk1e) in axons is interesting—can the authors share their thinking on this? The text states that the green nodes in this network are “highly upregulated in all 3 tip types”. What is the comparison group for this statement? (highly upregulated compared to what)

10) Figure 8 is hard to compare with Figure 7; the authors should state which up/downregulated genes present in the Venn diagrams were confirmed (and which not) by PCR.

11) In contrast to the gene expression data, IHC data are not well supported. First, the antibodies to csnk1e (a 47.3 kD protein) and map3k2 (a 78 kD protein) used in this study are not known to recognize the homologous lamprey proteins. Indeed, the Western blots show that each antibody recognizes a band at identical MW, which is between 50 and 60 kD. These do not seem likely to be csnk1e and map3k2; perhaps the bands represent an abundant protein such as tubulin? This means that the IHC performed with these antibodies will be difficult or impossible to interpret, even without the difficulties posed by differences between SDS-solubilized proteins and acetone-fixed tissue. In Figure 9, staining with csnk1e ab is in an odd pattern reminiscent of inclusion bodies, while the pattern of staining with map3k2 varies from light staining of the entire cell cytoplasm to strong staining of dense bodies. The light staining described for axons is not credible in the images shown (the use of sections incubated without primary antibody, which are not shown, controls only for non-specific binding of secondary antibodies). I don’t believe that this staining is interpretable in the current form. Figure 10 is similarly difficult to interpret, with the added issue that the location/extent of the axon tips stained in G through O of this figure cannot be determined from what is shown. In any case, unless the authors can provide data showing that these antibodies are recognizing csnk1e and map3k2 (respectively) in the tissue sections, I recommend that the antibody data be removed.

Author Response

Major comments:

This is a submittal from the laboratory most responsible for our understanding of spinal axon regeneration in the sea lamprey, and reflects the lab’s mastery of important techniques. In this manuscript, the authors describe an RNA-Seq analysis of gene expression in the tips of large lamprey spinal axons (large axons, not lampreys) in the subacute period following spinal cord transection. Some of these axons are destined to regenerate past the lesion site, while others will not. The unique contribution of this project is that, although the ultimate fate of these individual axon tips cannot be known, the authors could examine separately the genes expressed in axon tips that are advancing, retracting, or remaining static over a 4-hour period at a time (roughly 2 weeks after injury) immediately preceding RNA sampling from individual tips. This means that gene expression can be correlated with acute growth state in severed axons in vivo, which to my knowledge has not previously been accomplished.

 The authors analyze their gene expression data to determine differential expression of genes in each growth state compared to the other two, and use Gene Ontology analysis to suggest subcellular compartments and/or signaling pathways that appear relatively overrepresented in the three different groups. The gene expression data are well described and interesting, and the analysis suggests some hypotheses that could be tested in future work. There are, however, some issues with methodology and analysis that, in my opinion, should be addressed prior to acceptance for publication. Some of these are enumerated below.

Minor comments:

Q1: The authors need to provide the full RNASeq data sets as supplementary files; these data sets were not provided or not available for review. The value of this work will reside largely in the availability of these data according to FAIR data principles.

Absolutely correct.  We apologize for previously neglecting to provide this primary data, and now include it (Table S2) among 14 new items  of Supplemental Data

Q2: The authors do not describe the level(s) of the spinal cord from which the axon tips were collected. Were all tips collected from the same A-P region of the cord? Presumably this was proximal to the transection site, since it takes many weeks for axons to reach the lesion, but how far from the lesion site? Please provide this information.

Spinal cord transection was performed at the level of the 7th gill. Two weeks later, the spinal cord was exposed between the 3rd gill and the 7th gill (about 8 mm long) and any giant axon tips were imaged and aspirated. This is now described in more detail in Materials and Methods.

Q3: The impact of the work depends on the micrometer-level accuracy of measurements over the 4-hour sampling window. Therefore it would be helpful to have at least one example in which the “fiduciary marks” used for normalization can be seen in images from the two times investigated.

Figures 1 and 10 have been revised, with the areas containing the fiduciary landmarks encompassed within squares that have been adjusted for best brightness and contrast and pointed by pairs of arrowheads. Fiduciary landmarks are now clearer.

Q4: An FDR of 0.2 is not a common cutoff; the authors explain at one point that this was chosen so they could identify “proteins related to epigenetic modifications or…involved in signaling pathways”, but how a high FDR leads to this identification is not immediately obvious to me. Can this be clarified? In other places, FDRs of 0.05 or 0.1 are used; these choices should also be rationalized.

We now include a paragraph with more detailed explanation on page 11, “Results, DEG identification and DAVID enrichment analysis: “The different FDR values used above were selected to accomplish different goals. For DEGs analysis a low threshold (FDR <= 0.2) was selected in order to increase the chances of identifying genes whose encoded proteins are related to epigenetic modifications, or that are involved in signaling pathways, because these genes and coded proteins are more likely to be differentially expressed in different growth states, even if their expression levels are low. This increases the number of genes that could be analyzed by gene ontology (GO) enrichment, whereas a higher stringency criterion (FDR < 0.05) was used in most cases, in order to exclude noisy signals. The low-threshold FDR did not affect the identification of map3k2 as a gene present selectively in growing axon  tips.  Nor did it affect other analyses, such as Venn diagram analysis, or analyses of protein-protein interactions (PPI). GO terms with higher FDR values (FDR < 0.1 in Figure 5A, and Figure S3A; FDR < 0.2 in Figure 6A) are indicated and included only for comparison with corresponding scatter plots in Figures 5B, S3B, and 6B, respectively, where p values < 0.05 were used.

Q5: The data are said to be expressed as mean ± SD. If this is true, data like those in Figure 8 are unusual (p38beta is particularly striking). Can the authors confirm that SD was used? If so, I recommend that the authors instead plot each data point together with means and SEM or CV.

We apologize.  SD in Figure 8 is a mistake. We have corrected it to mean ± SEM. In addition, we have added the individual data points for each bar.

Q6: The authors should consider providing a clustering map for the 33 transcriptomes to show how the individual transcriptomes segregate (or not) with their groups. From Figure 3 it appears that this would not be completely consistent. This point relates to the PCA analysis, which indicates that the 3 groups have highly similar (indistinguishable) patterns of gene expression in the absence of two major outliers.

Three complete clustering maps have been provided (Figure S2A, 2B, and 2C).

Q7A: Selection of the 5 “most related transcriptomes” from each group was done in an attempt to reduce sample bias, but this selection has its own problems, since it reduces variability without objective criteria for deciding which tips are representative.

We now provide a Table S6 showing how we made the selection, based on correlation analysis.

Q7B: The authors then highlight a number of genes whose functions are partially understood (from GO terms) to make general suggestions about protein function in axon tips, which is an exercise that ought to be performed with caution.

The reviewer may be referring the gene lists in the Venn diagram. We listed all the results (genes) from GO enrichment analysis, no matter whether they are explainable or not.

Q7C: Why do the authors think that genes associated with TCR signaling would be upregulated in the R axon tips, for example? What is the importance of a putative circadian rhythm-related network in axon tips? GO analysis in Figure S1 suggests that differences between growing and static tips are mainly associated with nuclei. How are these findings interpreted? It would be helpful to know the authors’ views here.

We agree some data is unexpected or not understood. It results from the accuracy of the databases used for the GO analysis. However, some pathways may be shared by different cells or tissue, e.g., upregulation prkcq in R-tips. As indicated in Table S4 the prkcq gene is involved in more than one pathway, e.g., the keratinocyte pathway, the NF-kappa B signaling pathway, etc.. Just as in some cases, genes that are expressed in normal animals also have GO terms related to a disease.

Q8: On the other hand, there is already abundant evidence for the roles of ERKs, JNKs and p38s in axon growth and regeneration; to say that the Venn diagrams provided here suggest this idea seems to ignore a great deal of earlier work. This relates to an issue with the analysis in general; genes that authors choose to study are called “pivotal” while others with similar expression patterns are ignored. This approach seems to miss the opportunity provided by an unbiased screen of gene expression. It is recognized that there are choices that need to be made, but there should be some discussion of the pluses and minuses of this approach.

See above (reviewer 1, Q1A).

Q9A: In Figure S4, is there more than 1 grey rectangular node? If not, the legend should simply refer to spata32 in this context. Since spata32 protein is associated with microtubules and spermatogenesis, its presence at the center of a network supposedly related to circadian rhythms (and csnk1e) in axons is interesting—can the authors share their thinking on this? The text states that the green nodes in this network are “highly upregulated in all 3 tip types”. What is the comparison group for this statement? (highly upregulated compared to what)

Figure S4 is now Figure S7. We have changed the legend to “Gray rectangular node spata32 is important but not present in the sequencing lists.” Although spermatogenesis is regulated by circadian rhythms (Fusco et al., Front Endocrinol  12, 800693, 2021), there is no direct correlation between spata32 protein and circadian rhythm. In Discussion, we have added a section, “Csnk1e/circadian rhythm network in axon regeneration”, discussing the possible mechanisms by which csnk1e protein may be involved in axon regeneration. The spata32 gene is upregulated on day 32 in postnatal rat testis, and it is down-regulated in patients with teratozoospermia. IHC showed that the protein was expressed strongly in the sperm tails. Its function is unknown or may related to sperm motility because it is microtubule-associated. Lack of spata32 in lamprey axon tips may be due to incompleteness of genome annotation, because it is expressed only in the perikaryon, or because it is normally not expressed in neurons. If this gene is involved in axon regeneration, we would assume it is related to enhancement of axon transport or microtubule assembly.

Q9B: The text states that the green nodes in this network are “highly upregulated in all 3 tip types”. What is the comparison group for this statement? (highly upregulated compared to what)

We have modified the sentence as: “The csnk1e network consists of 24 nodes, among which 9 genes (green color) were highly upregulated in one of the 3 tip types. We provide 2 sections in Table S7, which present count data for the individual genes in these three tips.

Q10: Figure 8 is hard to compare with Figure 7; the authors should state which up/downregulated genes present in the Venn diagrams were confirmed (and which not) by PCR.

Genes selected for validation are not only from Figure 7, i.e., they are not only mapk pathway-related. We describe this in the manuscript:  “Based on results from DESeq2 and Venn diagram analysis, we selected several pivotal genes to validate their expression with q-PCR, including 5 conventional MAPKs (erk, jnk1/2, p38α/β), 2 genes limited to the v-R region of the Venn diagram (elp3, usp22), and 2 genes differentially expressed in G-tips (map3k2, csnk1e, G>S&R in Figure 3D).”

Q11: In contrast to the gene expression data, IHC data are not well supported. First, the antibodies to csnk1e (a 47.3 kD protein) and map3k2 (a 78 kD protein) used in this study are not known to recognize the homologous lamprey proteins. Indeed, the Western blots show that each antibody recognizes a band at identical MW, which is between 50 and 60 kD. These do not seem likely to be csnk1e and map3k2; perhaps the bands represent an abundant protein such as tubulin? This means that the IHC performed with these antibodies will be difficult or impossible to interpret, even without the difficulties posed by differences between SDS-solubilized proteins and acetone-fixed tissue. In Figure 9, staining with csnk1e ab is in an odd pattern reminiscent of inclusion bodies, while the pattern of staining with map3k2 varies from light staining of the entire cell cytoplasm to strong staining of dense bodies. The light staining described for axons is not credible in the images shown (the use of sections incubated without primary antibody, which are not shown, controls only for non-specific binding of secondary antibodies). I don’t believe that this staining is interpretable in the current form. Figure 10 is similarly difficult to interpret, with the added issue that the location/extent of the axon tips stained in G through O of this figure cannot be determined from what is shown. In any case, unless the authors can provide data showing that these antibodies are recognizing csnk1e and map3k2 (respectively) in the tissue sections, I recommend that the antibody data be removed.

We have repeated the Western blotting twice, using the same antibodies and found that each antibody labels one band in either brain or spinal cord. The band labeled by the map3k2 antibody is approximately 34 kDa, and the band labeled by the csnk1e antibody 44 kDa. We now provide even better Western blots with adjusted antibody concentrations.  In Figure 9, the staining patterns with the two antibodies are different, whether by chromogenic IHC, or by immunofluorescence. Therefore, we believe the images in Figure 10 are valid for both antibodies. To support the result of the Western blot, we have provided our original images in Figure S6.

Reviewer 3 Report

The manuscript entitled " Transcriptome of injured lamprey axon tips: Single cell RNA-seq suggests differential involvement of MAPK signaling pathways in axon retraction and regeneration after spinal cord injury takes a singular and unique approach to identify genes in neurites that may determine the regenerative status of individuals axons. This was possible due to the usage of a specific model organism lamprey and to the authors' accumulated experience in analyzing the axon regeneration in this specific model. It is a bit of regret that the results obtained in this interesting study were not fully validated, especially in a functional context, or lacked in-depth molecular analyses to pinpoint a handful of tempting targets that could be candidates for later functional analyses. However, the advantages of the unique approach taken in this study would outweigh the above shortcomings. Therefore, this manuscript will be a valuable addition to the growing list of sequencing studies in the field of axon regeneration. 

Author Response

Major comments:

……. It is a bit of regret that the results obtained in this interesting study were not fully validated, especially in a functional context, or lacked in-depth molecular analyses to pinpoint a handful of tempting targets that could be candidates for later functional analyses. However, the advantages of the unique approach taken in this study would outweigh the above shortcomings. Therefore, this manuscript will be a valuable addition to the growing list of sequencing studies in the field of axon

We agree with the reviewer.  We have done q-PCR and IHC for a few genes and have not attempted to do molecular manipulations yet, although in other studies we have used morpholino antisense oligonucleotide knockdown for a very limited number of genes. As explained above in response to Reviewer 1, although this report is only descriptive, it includes a great deal of valuable data that we and others can mine to make many more discoveries about the role of local protein synthesis in axon regeneration, and the roles of specific proteins in regeneration and degeneration.  Indeed, the lamprey may not be the best animal in which to test the functions of the discovered local mRNAs. Our data may point the way to experimental validations in species such as mice, which are more convenient for molecular manipulation, but in which it is very difficult to imagine obtaining the information contained in the present report.  This is now better explained in Conclusions.